**Evaluation of autoconversion and accretion enhancement factors in GCM warm-rain**
**parameterizations using ground-based measurements at the Azores**
Peng Wu[1], *Baike Xi[1], Xiquan Dong[1], and Zhibo Zhang[2]
[1] Department of Hydrology and Atmospheric Sciences, The University of Arizona, Tucson,
Arizona, USA
[2] Physics Department, The University of Maryland, Baltimore County, Maryland, USA
Submitted to Atmospheric Chemistry and Physics (November 21, 2018)
**Keywords**: MBL clouds, enhancement factors, autoconversion and accretion parameterizations

18 * Corresponding author address: Dr. Baike Xi, Department of Hydrology and Atmospheric

19 Sciences, University of Arizona, 1133 E. James E. Rogers Way, Tucson, AZ 85721-0011.

20 baike@email.arizona.edu; Phone: 520-626-8945

**Abstract**

A great challenge in climate modelling is how to parametrize sub-grid cloud processes, such as autoconversion and accretion in warm rain formation. In this study, we use ground-based observations and retrievals over the Azores to investigate the so-called enhancement factors, $E_{auto}$ and $E_{accr}$, which are often used in climate models to account for the influences of sub-grid variances of cloud and precipitation water on the autoconversion and accretion processes. $E_{auto}$ and $E_{accr}$ are computed for different equivalent model grid sizes. The calculated $E_{auto}$ values increases from 1.96 (30 km) to 3.2 (180 km), and the calculated $E_{accr}$ values increase from 1.53 (30 km) to 1.76 (180 km). Comparing the prescribed enhancement factors in Morrison and Gettleman (2008, MG08) to the observed ones, we found that a higher $E_{auto}$ (3.2) at small grids and lower $E_{accr}$ (1.07) are used in MG08, which helps to explain why most of the GCMs produce too frequent precipitation events but with too light precipitation intensity. The ratios of rain to cloud water mixing ratio at $E_{accr}$=1.07 and $E_{accr}$=2.0 are 0.063 and 0.142, respectively, from observations, further suggesting that the prescribed value of $E_{accr}$=1.07 used in MG08 is too small to simulate correct precipitation intensity. Both $E_{auto}$ and $E_{accr}$ increase when the boundary layer becomes less stable, and the values are larger in precipitating clouds (CLWP>75 gm$^{-2}$) than those in nonprecipiting clouds (CLWP<75 gm$^{-2}$). Therefore, the selection of $E_{auto}$ and $E_{accr}$ values in GCMs should be regime- and resolution- dependent.

**1. Introduction**

Due to their vast areal coverage (Warren et al., 1986, 1988; Hahn and Warren, 2007) and strong radiative cooling effect (Hartmann et al., 1992; Chen et al., 2000), small changes in the coverage or thickness of marine boundary layer (MBL) clouds could change the radiative energy budget significantly (Hartmann and Short, 1980; Randall et al., 1984) or even offset the radiative effects produced by increasing greenhouse gases (Slingo, 1990). The lifetime of MBL clouds remains an issue in climate models (Yoo and Li, 2012; Jiang et al., 2012; Yoo et al., 2013; Stanfield et al., 2014) and represents one of the largest uncertainties in predicting future climate (Wielicki et al., 1995; Houghton et al., 2001; Bony and Dufresne, 2005).

MBL clouds frequently produce precipitation, mostly in the form of drizzle (Austin et al., 1995; Wood, 2005a; Leon et al., 2008; Wood, 2012). A significant amount of drizzle is evaporated before reaching the surface, for example, about ~76% over the Azores region in Northeast Atlantic (Wu et al., 2015), which provides another water vapour source for MBL clouds. Due to their pristine environment and their close vicinity to the surface, MBL clouds and precipitation are especially sensitive to aerosol perturbations (Quaas et al., 2009; Kooperman et al., 2012). Thus, accurate prediction of precipitation is essential in simulating the global energy budget and in constraining aerosol indirect effects in climate projections.

Due to the coarse spatial resolutions of the general circulation model (GCM) grid, many cloud processes cannot be adequately resolved and must be parameterized. For example, warm

rain parameterizations in most GCMs treat the condensed water as either cloud or rain from the
collision-coalescence process that is partitioned into autoconversion and accretion sub-
processes in model parameterizations (Kessler, 1969; Tripoli and Cotton, 1980; Beheng, 1994;
Khairoutdinov and Kogan, 2000; Liu and Daum, 2004). Autoconversion represents the process
that drizzle drops being formed through the condensation of cloud droplets and accretion
represents the process where rain drops grow by the coalescence of drizzle-sized drops with
cloud droplets. Autoconversion mainly accounts for precipitation initiation while accretion
primarily contributes to precipitation intensity. Autoconversion is often parameterized as
functions of cloud droplet number concentration ($N_c$) and cloud water mixing ratio ($q_c$), while
accretion depends on both cloud and rain water mixing ratios ($q_c$ and $q_r$) (Kessler, 1969; Tripoli
and Cotton, 1980; Beheng, 1994; Khairoutdinov and Kogan, 2000; Liu and Daum, 2004;
Wood, 2005b). The majority of previous studies suggested that these two processes as power
law functions of cloud and precipitation properties (See section 2 for details).

In conventional GCMs, the lack of information on the sub-grid variances of cloud and

precipitation leads to the unavoidable use of the grid-mean quantities ($\overline{N_c}$, $\overline{q_c}$, and $\overline{q_r}$, where
overbar denotes grid mean, same below) in calculating autoconversion and accretion rates.
MBL cloud liquid water path (CLWP) distributions are often positive skewed (Wood and
Hartmann, 2006; Dong et al., 2014a and 2014b), that is, the mean value is greater than mode
value. Thus, the mean value only represents a relatively small portion of samples. Also, due to
the nonlinear nature of the relationships, the two processes depend significantly on the sub-
grid variability and co-variability of cloud and precipitation microphysical properties (Weber
and Quass, 2012; Boutle et al., 2014). In some GCMs, sub-grid scale variability is often ignored
or hard coded using constants to represent the variabilities under all meteorological conditions
and across the entire globe (Pincus and Klein, 2000; Morrison and Gettleman, 2008; Lebsock
et al., 2013). This could lead to systematic errors in precipitation rate simulations (Wood et al.,
2002; Larson et al., 2011; Lebsock et al., 2013; Boutle et al., 2014; Song et al., 2018), where
GCMs are found to produce too frequent but too light precipitation compared to observations
(Zhang et al., 2002; Jess, 2010; Stephens et al., 2010; Nam and Quaas, 2012; Song et al., 2018).
The bias is found to be smaller by using a probability density function (PDF) of cloud water to
represent the sub-grid scale variability in autoconversion parameterization (Beheng, 1994;
Zhang et al., 2002; Jess, 2010), or more complexly, by integrating the autoconversion rate over
a joint PDF of liquid water potential temperature, and total water mixing ratio (Cheng and Xu,

2009).

Process rate enhancement factors ($E$) are introduced when considering sub-grid scale
variability in parameterizing grid-mean processes and they should be parameterized as
functions of the PDFs of cloud and precipitation properties within a grid box (Morrison and
Gettleman, 2008; Lebsock et al., 2013; Boutle et al., 2014). However, these values in some
GCM parameterization schemes are prescribed as constants regardless of underlying surface

or meteorological conditions (Xie and Zhang, 2015). Boutle et al. (2014) used aircraft in situ measurements and remote sensing techniques to develop a parameterization for cloud and rain, in which not only consider the sub-grid variabilities under different grid scales, but also consider the variation of cloud and rain fractions. The parameterization was found to reduce precipitation estimation bias significantly. Hill et al. (2015) modified this parameterization and developed a regime and cloud type dependent sub-grid parameterization, which was implemented to the Met Office Unified Model by Walters et al. (2017) and found that the radiation bias is reduced using the modified parameterization. Using ground-based observations and retrievals, Xie and Zhang (2015) proposed a scale-aware cloud inhomogeneity parameterization that they applied to the Community Earth System Model (CESM) and found that it can recognize spatial scales without manual tuning and can be applied to the entire globe. The inhomogeneity parameter is essential in calculating enhancement factors and affect the conversion rate from cloud to rain liquid. Xie and Zhang (2015), however, did not evaluate the validity of CESM simulations from their parameterization; the effect of $N_c$ variability or the effect of covariance of cloud and rain on accretion process was not assessed. Most recently, Zhang et al. (2018) derived the sub-grid distribution of CLWP and $N_c$ from the MODIS cloud product. They also studied the implication of the sub-grid cloud property variations for the autoconversion rate simulation, in particular the enhancement factor, in GCMs. For the first time, the enhancement factor due to the sub-grid variation of $N_c$ is derived

from satellite observation, and results reveal several regions downwind of biomass burning
aerosols (e.g., Gulf of Guinea, East Coast of South Africa), air pollution (i.e., Eastern China
Sea), and active volcanos (e.g., Kilauea Hawaii and Ambae Vanuatu), where the enhancement
factor due to $N_c$ is comparable, or even larger than that due to CLWP. However, one limitation
of Zhang et al. (2018) is the use of passive remote sensing data only, which cannot distinguish
cloud and rain water.
Dong et al. (2014a and 2014b) and Wu et al. (2015) reported MBL cloud and rain properties
over the Azores and provided the possibility of calculating the enhancement factors using
ground-based observations and retrievals. A joint retrieval method to estimate $q_c$ and $q_r$ profiles
is proposed based on existing studies and is presented in Appendix A. Most of the calculations
and analyses in this study is based on Morrison and Gettleman (2008, MG08 hereafter) scheme.
The enhancement factors in several other schemes are also discussed and compared with the
observational results and the approach in this study can be repeated for other microphysics
schemes in GCMs. This manuscript is organized as follows: section 2 includes a summary of
the mathematical formulas from previous studies that can be used to calculate enhancement
factors. Ground-based observations and retrievals are introduced in Section 3. Section 4
presents results and discussions, followed by summary and conclusions in Section 5. The
retrieval method used in this study is in Appendix A.

## 2. Mathematical Background

Autoconversion and accretion rates in GCMs are usually parameterized as power law equations (Tripoli and Cotton, 1980; Beheng, 1994; Khairoutdinov and Kogan, 2000; Liu and Daum, 2004):

$$\left(\frac{\partial q_r}{\partial t}\right)_{auto} = A\overline{q_c}^{a1}\overline{N_c}^{-a2}, \tag{1}$$

$$\left(\frac{\partial q_r}{\partial t}\right)_{accr} = B(\overline{q_c}\,\overline{q_r})^b, \tag{2}$$

where $A$, $a1$, $a2$, $B$, and $b$ are coefficients in different schemes listed in Table 1. The $\overline{q_c}$, $\overline{q_r}$, and $\overline{N_c}$ are grid-mean cloud water mixing ratio, rain water mixing ratio, and droplet number concentration, respectively. Because it is widely used in model parameterizations, the detailed results from Khairoutdinov and Kogan (2000) parameterization that been used in MG08 scheme will be shown in Section 4 while a summary will be given for other schemes.

Ideally, the covariance between physical quantities should be considered in the calculation of both processes. However, $\overline{q_c}$ and $\overline{N_c}$ in Eq. (1) are arguably not independently retrieved in our retrieval method which will be introduced in this section and Appendix A. Thus we only assess the individual roles of $q_c$ and $N_c$ sub-grid variations in determining autoconversion rate. $q_c$ and $q_r$, on the other hand, are retrieved from two independent algorithms as shown in Dong et al. (2014a and 2014b), Wu et al. (2015) and Appendix A, we will assess the effect of cloud and rain property covariance on accretion rate calculations.

In the sub-grid scale, the PDFs of $q_c$ and $N_c$ are assumed to follow a gamma distribution
based on observational studies of optical depth in MBL clouds (Barker et al., 1996; Pincus et
al., 1999; Wood and Hartmann, 2006):
$$P(x) = \frac{\alpha^\nu}{\Gamma(\nu)} x^{\nu-1} e^{-\alpha x} ,$$    (3)
where $x$ represents $q_c$ or $N_c$ with grid-mean quantity $\overline{q_c}$ or $\overline{N_c}$, represented by $\mu$, $\alpha = \nu/\mu$ is the
scale parameter, $\sigma^2$ is the relative variance of $x$ (= variance divided by $\mu^2$), $\nu = 1/\sigma^2$ is the
shape parameter. $\nu$ is an indicator of cloud field homogeneity, with large values representing
homogeneous and small values indicating inhomogeneous cloud field.
By integrating autoconversion rate, Eq. (1), over the grid-mean rate, Eq. (3), with respect
to sub-grid scale variation of $q_c$ and $N_c$, the autoconversion rate can be expressed as:
$$\left(\frac{\partial q_r}{\partial t}\right)_{auto} = A\mu_{q_c}^{a1} \mu_{N_c}^{a2} \frac{\Gamma(\nu+a)}{\Gamma(\nu)\nu^a},$$    (4)
where $a = a1$ or $a2$. Comparing Eq. (4) to Eq. (1), the autoconversion enhancement factor
($E_{auto}$) can be given with respect to $q_c$ and $N_c$:
$$E_{auto} = \frac{\Gamma(\nu+a)}{\Gamma(\nu)\nu^a} .$$    (5)
In addition to fitting the distributions of $q_c$ and $N_c$, we also tried two other methods to
calculate $E_{auto}$. The first is to integrate Eq. (1) over the actual PDFs from observed or retrieved
parameters and the second is to fit a lognormal distribution for sub-grid variability like what
has been done in other studies (e.g., Lebsock et al., 2013; Larson and Griffin, 2013). It is found
that all three methods get similar results. In this study, we use a gamma distribution that is
consistent with MG08. Also note that, in the calculation of $E_{auto}$ from $\overline{N_c}$, the negative exponent
(-1.79) may cause singularity problems in Eq. (5). When this situation occurs, we do direct
calculations by integrating the PDF of $\overline{N_c}$ rather than using Eq. (5).
To account for the covariance of microphysical quantities in a model grid, it is difficult to
apply bivariate gamma distribution due to its complex nature. In this study, the bivariate
lognormal distribution of $q_c$ and $q_r$ is used (Lebsock et al., 2013; Boutle et al., 2014) and can
be written as:
$P(\overline{q_c},\ \overline{q_r}) = \frac{1}{2\pi \overline{q_c}\ \overline{q_r}\sigma_{q_c}\sigma_{q_r}\sqrt{1-\rho^2}} exp\left\{-\frac{1}{2}\frac{1}{1-\rho^2}\left[\left(\frac{ln\overline{q_c}-\mu_{q_c}}{\sigma_{q_c}}\right)^2 - 2\rho\left(\frac{ln\overline{q_c}-\mu_{q_c}}{\sigma_{q_c}}\right)\left(\frac{ln\overline{q_r}-\mu_{q_r}}{\sigma_{q_r}}\right)+\right.\right.$
$\left.\left.\left(\frac{ln\overline{q_r}-\mu_{q_r}}{\sigma_{q_r}}\right)^2\right]\right\},$       (6)
where $\sigma$ is standard deviation and $\rho$ is the correlation coefficient of $q_c$ and $q_r$.
Similarly, by integrating the accretion rate in Eq. (2) from Eq. (6), we get the accretion
enhancement factor ($E_{accr}$) of:
$E_{accr} = \left(1+\frac{1}{v_{q_c}}\right)^{\frac{1.15^2-1.15}{2}}\left(1+\frac{1}{v_{q_r}}\right)^{\frac{1.15^2-1.15}{2}}\exp(\rho 1.15^2\sqrt{\ln\left(1+\frac{1}{v_{q_c}}\right)\ln(1+\frac{1}{v_{q_r}})}).$    (7)

## 3. Ground-based observations and retrievals

The datasets used in this study were collected at the Department of Energy (DOE) Atmospheric Radiation Measurement (ARM) Mobile Facility (AMF), which was deployed on the northern coast of Graciosa Island (39.09°N, 28.03°W) from June 2009 to December 2010 (for more details, please refer to Rémillard et al., 2012; Dong et al., 2014a and Wood et al., 2015). The detailed operational status of the remote sensing instruments on AMF was summarized in Figure 1 of Rémillard et al. (2012) and discussed in Wood et al. (2015). The ARM Eastern North Atlantic (ENA) site was established on the same island in 2013 and provides long-term continuous observations.

The cloud-top heights ($Z_{top}$) were determined from W-band ARM cloud radar (WACR) reflectivity and only single-layered low-level clouds with $Z_{top} \leq 3$ km are selected. Cloud-base heights ($Z_{base}$) were detected by a laser ceilometer (CEIL) and the cloud thickness was simply the difference between cloud top and base heights. The cloud liquid water path (CLWP) was retrieved from microwave radiometer (MWR) brightness temperatures measured at 23.8 and 31.4 GHz using a statistical retrieval method with an uncertainty of 20 g m$^{-2}$ for CLWP < 200 g m$^{-2}$, and 10% for CLWP > 200 g m$^{-2}$ (Liljegren et al., 2001; Dong et al., 2000). Precipitating status is identified through a combination of WACR reflectivity and $Z_{base}$. As in Wu et al. (2015), we labelled the status of a specific time as "precipitating" if the WACR reflectivity below the cloud base exceeds -37 dBZ.

The ARM merged sounding data have a 1-min temporal and 20-m vertical resolution below
3 km (Troyan, 2012). In this study, the merged sounding profiles are averaged to 5-min
resolution. Pressure and temperature profiles are used to calculate air density ($\rho_{air}$) profiles
and to infer adiabatic cloud water content.
Cloud droplet number concentration ($N_c$) is retrieved using the methods presented in Dong
et al. (1998, 2014a and 2014b) and are assumed to be constant in a cloud layer. Vertical profiles
of cloud and rain water content (CLWC and RLWC) are retrieved by combining WACR
reflectivity, CEIL attenuated backscatter and by assuming adiabatic growth of cloud parcels.
The detailed description is presented in Appendix A with the results from a selected case. The
CLWC and RLWC values are transformed to $q_c$ and $q_r$ by dividing by air density (e.g., $q_c(z) =$
$CLWC(z)/\rho_{air}(z)$).
The estimated uncertainties for the retrieved $q_c$ and $q_r$ are 30% and 18%, respectively (see
Appendix A). We used the estimated uncertainties of $q_r$ and $q_c$ as inputs of Eqs. (4) and (7) to
assess the uncertainties of $E_{auto}$ and $E_{accr}$. For instance, $(1 \pm 0.3)q_c$ are used in Eq. (4) and the
mean differences are then used as the uncertainty of $E_{auto}$. Same method is used to estimate the
uncertainty for $E_{accr}$.
The autoconversion and accretion parameterizations partitioned from the collision-
coalescence process dominate at different levels in a cloud layer. Autoconversion dominates
around cloud top where cloud droplets reach maximum by condensation and accretion is
dominant at middle and lower parts of the cloud where rain drops sediment and continue to
grow by collecting cloud droplets. Complying with the physical processes, we estimate
autoconversion and accretion rates at different levels of a cloud layer in this study. The
averaged $q_c$ within the top five range gates (~215 m thick) are used to calculate $E_{auto}$. To
calculate $E_{accr}$, we use the averaged $q_c$ and $q_r$ within five range gates around the maximum
radar reflectivity. If the maximum radar reflectivity appears at the cloud base, then five range
gates above the cloud base are used.
The ARM merged sounding data are also used to calculate lower tropospheric stability
($LTS = \theta_{700\,hPa} - \theta_{1000\,hPa}$), which is used to infer the boundary layer stability. In this study,
unstable and stable boundary layers are defined as LTS less than 13.5 K and greater than 18 K,
respectively, and environment with an LTS between 13.5 K and 18 K is defined as mid-stable
(Wang et al. 2012; Bai et al. 2018). Enhancement factors in different boundary layers are
summarized in Section 4.2 and may be used as references for model simulations. Further, two
regimes are classified: CLWP greater than 75 g m$^{-2}$ as precipitating and CLWP less than 75 g
m$^{-2}$ as nonprecipitating (Rémillard et al., 2012).
To evaluate the dependence of autoconversion and accretion rates on sub-grid variabilities
for different model spatial resolutions, an averaged wind speed within a cloud layer was
extracted from merged sounding and used in sampling observations over certain periods to
mimic different grid sizes in GCMs. For example, two hours of observations corresponds to a
72-km horizontal equivalent grid box if mean in-cloud wind speed is 10 $m\ s^{-1}$ horizontal wind
and if the wind speed is 5 $m\ s^{-1}$, four hours of observations is needed to mimic the same
horizontal equivalent grid. We used six horizontal equivalent grid sizes (30-, 60-, 90-, 120-,
150-, and 180-km) and mainly show the results from 60-km and 180-km horizontal equivalent
grid sizes in Section 4. For convenience, we refer 'equivalent size' as 'horizontal equivalent
grid size' from now on.
**4. Results and discussions**
In this section, we first show the data and methods using a selected case, followed by
statistical analysis based on 19 months of data and multiple time-intervals.
**4.1 Case study**
The selected case occurred on July 27, 2010 (Figure 1a) at the Azores. This case was
characterized by a long time of non-precipitating or light drizzling cloud development (00:00-
14:00 UTC) before intense drizzling occurred (14:00-20:00 UTC). Wu et al. (2017) studied
this case in detail to demonstrate the effect of wind shear on drizzle initiation. Here, we choose
two periods corresponding to a 180-km equivalent size and having similar mean $q_c$ near cloud
top: 0.28 g kg$^{-1}$ for period c and 0.26 g kg$^{-1}$ for period d but with different distributions (Figures
1c and 1d). The PDFs of $q_c$ are then fitted using gamma distributions to get shape parameters
($\nu$) as shown in Figures 1c and 1d. Smaller $\nu$ is usually associated with a more inhomogeneous
cloud field, which allows more rapid drizzle production and more efficient liquid
transformation from cloud to rain (Xie and Zhang, 2015) in regions that satisfy precipitation
criteria, which is usually controlled using threshold $q_r$, droplet size or relative humidity
(Kessler, 1969; Liu and Daum, 2004). The period d has a wider $q_c$ distribution than the period
c, resulting in a smaller $\nu$ and thus larger $E_{auto}$. Using the fitted $\nu$, the $E_{auto}$ from $q_c$ is calculated
from Eq. (5) and the period d is larger than the period c (1.80 vs. 1.33). The $E_{auto}$ values for the
periods d and c can also be calculated from $N_c$ using the same procedure as $q_c$ with a similar
result (2.1 vs. 1.51). The $E_{accr}$ values for the periods d and c can be calculated from the
covariance of $q_c$ and $q_r$ and Eq. (7).  Not surprisingly, the period d has larger $E_{accr}$ than the
period c. The combination of larger $E_{auto}$ and $E_{accr}$ in the period d contributes to the rapid drizzle
production and high rain rate as seen from WACR reflectivity and $q_r$ in Figure A1.

It is important to understand the physical meaning of enhancement factors in precipitation

parameterization. For example, if we assume two scenarios for $q_c$ with a model grid having the
same mean values but different distributions: (1) The distribution is extremely homogeneous,
there will be no sub-grid variability because the cloud has the same chance to precipitate and
the enhancement factors would be unity (this is true for arbitrary grid-mean $q_c$ amount as well).
(2) The cloud field gets more and more inhomogeneous with a broad range of $q_c$ within the
model grid box, which results in a greater enhancement factor and increases the possibility of
precipitation. That is, a large enhancement factor can make the part of the cloud with higher $q_c$
within the grid box become more efficient in generating precipitation, rather than the entire
model grid.
Using the LWP retrieved from the Moderate Resolution Imaging Spectroradiometer
(MODIS) as an indicator of cloud inhomogeneous, Wood and Hartmann (2006) found that
when clouds become more inhomogeneous, cloud fraction decreases, and open cells become
dominant with stronger drizzling process (Comstock et al., 2007). The relationship between
reduced homogeneity and stronger precipitation intensity is found in this study, which is similar
to the findings in other studies (e.g., Wood and Hartmann, 2006, Comstock et., 2007, Barker
et al., 1996; Pincus et al., 1999).
It is clear that $q_c$ and $N_c$ in Figure 1b are correlated with each other. In addition to their
natural relationships, $q_c$ and $N_c$ in our retrieval method are also correlated (Dong et al., 2014a
and 2014b). Thus, the effect of $q_c$ and $N_c$ covariance on $E_{auto}$ is not included in this study. In
Figures 1c and 1d, the results are calculated using equivalent size of 180-km for the selected
case on 27 July 2010. In Section 4.2, we will use these approaches to calculate their statistical
results for multiple equivalent sizes using the 19-month ARM ground-based observations and
retrievals.
**4.2 Statistical result**
For a specific equivalent size, e.g. 60-km, we estimate the shape parameter ($\nu$) and calculate
$E_{auto}$ through Eqns. (5) and (7). The PDFs of $E_{auto}$ for both 60-km and 180-km equivalent sizes
are shown in Figures 2a-2d. The distributions of $E_{auto}$ values calculated from $q_c$ with 60-km
and 180-km equivalent sizes (Figures 2a and 2b) are different to each other (2.79 vs. 3.3). The
calculated $E_{auto}$ values range from 1 to 10, and most are less than 4. The average value for the
60-km equivalent size (2.79) is smaller than that for the 180-km equivalent size (3.2), indicating
a possible dependence of $E_{auto}$ on model grid size. Because drizzle-sized drops are primarily
resulted from the autoconversion, we investigate the relationship between $E_{auto}$ and
precipitation frequency, which is defined as the average percentage of drizzling occurrence
based on radar reflectivity below the cloud base. Given the average LWP at Azores from Dong
et al. (2014b, 109-140 g m$^{-2}$), the precipitation frequency (black lines in Figures 2a and 2b)
agrees well with those from Kubar et al. (2009, 0.1-0.7 from their Figure 11). The precipitation
frequency within each bin shows an increasing trend for $E_{auto}$ from 0 to 4-6, then oscillates
when $E_{auto} > 6$, indicating that in precipitation initiation process, $E_{auto}$ keeps increasing to a
certain value (~6) until the precipitation frequency reaches a near-steady state. Larger $E_{auto}$
values do not necessarily result in higher precipitation frequency but instead may produce more
drizzle-sized drops from autoconversion process when the cloud is precipitating.
The PDFs of $E_{auto}$ calculated from $N_c$ also share similar patterns of positive skewness and
peaks at ~1.5-2.0 for the 60-km and 180-km equivalent sizes (Figures 2c and 2d). Although the
average values are close to their $q_c$ counterparts (2.54 vs. 2.79 for 60-km and 3.45 vs. 3.2 for
180-km), the difference in $E_{auto}$ between 60-km and 180-km equivalent sizes becomes large.
The precipitation frequencies within each bin are nearly constant or slightly decrease, which
are different to their $q_c$ counterparts shown in Figures 2a and 2b. This suggests complicated
effects of droplet number concentration on precipitation initiation and warrants more
explorations of aerosol-cloud-precipitation interactions. As mentioned in Section 2, $q_c$ and $N_c$
are also fitted using lognormal distributions to calculate $E_{auto}$, those are close to the results in
Figure 2 (not shown here) with average values of 3.28 and 3.84, respectively, for 60-km and
180-km equivalent sizes. Because the $E_{auto}$ values calculated from $q_c$ and $N_c$ are close to each
other, we will focus on analyzing the results from $q_c$ only for simplicity and clarity. The effect
of $q_c$ and $N_c$ covariance, as stated in Section 4.1, is not presented in this study due to the intrinsic
correlation in the retrieval (Dong et al., 2014a and 2014b and Appendix A of this study).
The covariance of $q_c$ and $q_r$ is included in calculating $E_{accr}$ and the results are shown in
Figures 2e and 2f. The calculated $E_{accr}$ values range from 1 to 4 with mean values of 1.62 and
1.76 for 60-km and 180-km equivalent sizes, respectively. These two mean values are much
greater than the prescribed value used in MG08 (1.07). Since accretion is dominant at middle
and lower parts of the cloud where rain drops sediment and continue to grow by collecting
cloud droplets, we superimpose the ratio of $q_r$ to $q_c$ within each bin (black lines in Figures 2e
and 2f) to represent the portion of rain water in the cloud layer. In both panels, the ratios are
less than 15%, which means that $q_r$ can be one order of magnitude smaller than $q_c$. The
differences in magnitude are consistent with previous CloudSat and aircraft results (e.g., Boutle
et al., 2014). This ratio increases from $E_{accr}$=0 to ~2, and then decreases, suggesting a possible
optimal state for the collision-coalescence process to achieve maximum efficiency for
converting cloud water into rain water at $E_{accr}$=2. In other words, the conversion efficiency
cannot be infinitely increased with $E_{accr}$ under available cloud water. The ratio of $q_r$ to $q_c$
increases from $E_{accr}$=1.07 (0.063) to $E_{accr}$=2.0 (0.142), indicating that the fraction of rain water
in total water using the prescribed $E_{accr}$ is too low.  This ratio could be increased significantly
using a large $E_{accr}$ value, therefore increasing precipitation intensity in the models. This further
proves that the prescribed value of $E_{accr}$=1.07 used in MG08 is too small to correctly simulate
precipitation intensity in the models. Therefore, similar to the conclusions in Lebsock et al.
(2013) and Boutle et al. (2014), we suggest increasing $E_{accr}$ from 1.07 to 1.5-2.0 in GCMs.

To illustrate the impact of using prescribed enhancement factors, autoconversion and

accretion rates are calculated using the prescribed values (e.g., 3.2 for $E_{auto}$ and 1.07 for $E_{accr}$,
MG08; Xie and Zhang, 2015) and the newly calculated ones in Figure 2 that use observations
and retrievals. Figure 3 shows the joint density of autoconversion (Figures 3a and 3b) and
accretion rates (Figures 3c and 3d) from observations (x-axis) and model parameterizations (y-
axis) for 60-km and 180-km equivalent sizes. Despite the spread, the peaks of the joint density
of autoconversion rate appear slightly above the one-to-one line especially for the 60-km
equivalent size, suggesting that cloud droplets in the model are more easily to be converted
into drizzle/rain drops than observations. On the other hand, the peaks of accretion rate appear
slightly below the one-to-one line which indicates that simulated precipitation intensities are
lower than observed ones. The magnitudes of the two rates are consistent with Khairoutdinov
and Kogan (2000), Liu and Daum (2004), and Wood (2005b).
Compared to the observations, the precipitation in GCMs occurs at higher frequencies with
lower intensities, which might explain why the total precipitation amounts are close to surface
measurements over an entire grid box. This 'promising' result, however, fails to simulate
precipitation on the right scale and cannot capture the correct rain water amount, thus providing
limited information in estimating rain water evaporation and air-sea energy exchange.
Clouds in an unstable boundary layer have a better chance of getting moisture supply from
the surface by upward motion than clouds in a stable boundary layer. Precipitation frequencies
are thus different in these two boundary layer regimes. For example, clouds in a relatively
unstable boundary layer more easily produce drizzle than those in a stable boundary layer (Wu
et al., 2017). Provided the same boundary layer condition, CLWP is an important factor in
determining the precipitation status of clouds. At the Azores, precipitating clouds are more
likely to have CLWP greater than 75 g m$^{-2}$ than their nonprecipitating counterparts (Rémillard
et al., 2012). To further investigate what conditions and parameters can significantly influence
the enhancement factors, we classify low-level clouds according to their boundary layer
conditions and CLWPs.
The averaged $E_{auto}$ and $E_{accr}$ values for each category are listed in Table 2. Both $E_{auto}$ and
$E_{accr}$ increase when the boundary layer becomes less stable, and these values become larger in
precipitating clouds (CLWP>75 gm$^{-2}$) than those in nonprecipiting clouds (CLWP<75 gm$^{-2}$).
In real applications, autoconversion process only occurs when $q_c$ or cloud droplet size reaches
a certain threshold (e.g., Kessler, 1969 and Liu and Daum, 2004). Thus, it will not affect model
simulations if a valid $E_{auto}$ is assigned to Eq. (1) in a nonprecipitating cloud. The $E_{auto}$ values
in both stable and mid-stable boundary layer conditions are smaller than the prescribed value
of 3.2, while the values in unstable boundary layers are significantly larger than 3.2 regardless
of if they are precipitating or not. All $E_{accr}$ values are greater than the constant of 1.07. The
$E_{auto}$ values in Table 2 range from 2.32 to 6.94 and the $E_{accr}$ values vary from 1.42 to 1.86,
depending on different boundary layer conditions and CLWPs. Therefore, as suggested by Hill
et al. (2015), the selection of $E_{auto}$ and $E_{accr}$ values in GCMs should be regime-dependent.
To properly parameterize sub-grid variabilities, the approaches by Hill et al. (2015) and
Walters et al. (2017) can be adopted. To use MG08 and other parameterizations in GCMs as
listed in Table 1, proper adjustments can be made according to the model grid size, boundary
layer conditions, and precipitating status. As stated in the methodology, we used a variety of
equivalent sizes. Figure 4 demonstrates the dependence of both enhancement factors on
different model grid sizes. The $E_{auto}$ values (red line) increase from 1.97 at an equivalent size
of 30 km to 3.15 at an equivalent size of 120 km, which are 38.4% and 2% percent lower than
the prescribed value (3.2, upper dashed line). After that, the $E_{auto}$ values remain relatively
constant of ~3.18 when the equivalent model size is 180 km, which is close to the prescribed
value of 3.2 used in MG08. This result indicates that the prescribed value in MG08 represents
well in large grid sizes in GCMs. The $E_{accr}$ values (blue line) increase from 1.53 at an equivalent
size of 30 km to 1.76 at an equivalent size of 180 km, those are 43% and 64%, respectively,
larger than the prescribed value (1.07, lower dashed line). The shaded areas represent the
uncertainties of $E_{auto}$ and $E_{accr}$ associated with the uncertainties of the retrieved $q_c$ and $q_r$. When
equivalent size increases, the uncertainties slightly decrease. The prescribed $E_{auto}$ is close to
the upper boundary of uncertainties except for the 30-km equivalent size, while the prescribed
$E_{accr}$ is significantly lower than the lower boundary.
It is noted that $E_{auto}$ and $E_{accr}$ depart from their prescribed values at opposite directions as
the equivalent size increases. For models with finer resolutions (e.g., 30-km), both $E_{auto}$ and
$E_{accr}$ are significantly different from the prescribed values, which can partially explain the issue
of 'too frequent' and 'too light' precipitation. Under both conditions, the accuracy of
precipitation estimation is degraded. For models with coarser resolutions (e.g., 180-km),
average $E_{auto}$ is exactly 3.2 while $E_{accr}$ is much larger than 1.07 when compared to finer
resolution simulations. In such situations, the simulated precipitation will be dominated by the
'too light' problem, in addition to regime-dependent (Table 2) and as in Xie and Zhang (2015),
$E_{auto}$ and $E_{accr}$ should be also scale-dependent.
Also note that the location of ground-based observations and retrievals used in this study is
on the remote ocean where the MBL clouds mainly form in a relatively stable boundary layer
and are characterized by high precipitation frequency. Even in such environments, however,
the GCMs overestimate the precipitation frequency (Ahlgrimm and Forbes, 2014).
To further investigate how enhancement factors affect precipitation simulations, we use
$E_{auto}$ as a fixed value of 3.2 in Eq. (4), and then calculate the $q_c$ needed for models to reach the
same autoconversion rate as observations. The $q_c$ differences between models and observations
are then calculated, which represent the $q_c$ adjustment in models to get a realistic
autoconversion rate in the simulations. Similar to Figure 1, the PDFs of $q_c$ differences (model
– observation) are plotted in Figures 5a and 5b for 60-km and 180-km equivalent sizes. Figure
5c shows the average percentages of model $q_c$ adjustments for different equivalent sizes. The
mode and average values for 30-km equivalent size is negative, suggesting that models need to
simulate lower $q_c$ in general to get reasonable autoconversion rates. Lower $q_c$ values are usually
associated with smaller $E_{auto}$ values that induce lower simulated precipitation frequency. On
average, the percentage of $q_c$ adjustments decrease with increasing equivalent size. For
example, the adjustments for finer resolutions (e.g., 30-60 km) can be ~20% of the $q_c$, whereas
adjustments in coarse resolution models (e.g., 120 – 180 km) are relatively small because the
prescribed $E_{auto}$ (=3.2) is close to the observed ones (Figure 4) and when equivalent size is 180-
km, no adjustment is needed. The adjustment method presented in Figure 5, however, may
change cloud water substantially and may cause a variety of subsequent issues, such as altering
cloud radiative effects and disrupting the hydrological cycle. The assessment in Figure 5 only
provides a reference to the equivalent effect on cloud water by using the prescribed $E_{auto}$ value
as compared to those from observations.
All above discussions are based on the prescribed $E_{auto}$ and $E_{accr}$ values (3.2 and 1.07) in
MG08. Whereas there are quite a few parameterizations that have been published so far. In this
study, we list $E_{auto}$ and $E_{accr}$ for three other widely used parameterization schemes in Table 3,
which are given only for 60-km and 180-km equivalent sizes. The values of the exponent in
each scheme directly affect the values of the enhancement factors. For example, the scheme in
Beheng (1994) has highest degree of nonlinearity and hence has the largest enhancement
factors. The scheme in Liu and Daum (2004) is very similar to the scheme in Khairoutdinov
and Kogan (2000) because both schemes have a physically realistic dependence on cloud water
content and number concentration (Wood, 2005b). For a detailed overview and discussion of
various existing parameterizations, please refer to Liu and Daum (2004), Liu et al. (2006a), Liu
et al. (2004b), Wood (2005b) and Michibata and Takemura (2015). A physical based
autoconversion parameterization was developed by Lee and Baik (2017) in which the scheme
was derived by solving stochastic collection equation with an approximated collection kernel
that is constructed using the terminal velocity of cloud droplets and the collision efficiency
obtained from a particle trajectory model. Due to the greatly increased complexity of their
equation, we do not attempt to calculate $E_{auto}$ here but should be examined in future studies due
to the physics feasibility of the Lee and Baik (2017) scheme.

**5. Summary**
To better understand the influence of sub-grid cloud variations on the warm-rain process
simulations in GCMs, we investigated the warm-rain parameterizations of autoconversion
($E_{auto}$) and accretion ($E_{accr}$) enhancement factors in MG08. These two factors represent the
effects of sub-grid cloud and precipitation variabilities when parameterizing autoconversion
and accretion rates as functions of grid-mean quantities. $E_{auto}$ and $E_{accr}$ are prescribed as 3.2
and 1.07, respectively, in the widely used MG08 scheme. To assess the dependence of the two
parameters on sub-grid scale variabilities, we used ground-based observations and retrievals
collected at the DOE ARM Azores site to reconstruct the two enhancement factors in different
equivalent sizes.
From the retrieved $q_c$ and $q_r$ profiles, the averaged $q_c$ within the top five range gates are
used to calculate $E_{auto}$ and the averaged $q_c$ and $q_r$ within five range gates around maximum
reflectivity are used to calculate $E_{accr}$. The calculated $E_{auto}$ values from observations and
retrievals increase from 1.96 at an equivalent size of 30 km to 3.18 at an equivalent size of 150
km. These values are 38% and 0.625% lower than the prescribed value of 3.2. The prescribed
value in MG08 represents well in large grid sizes in GCMs (e.g., $180^2$ km$^2$ grid). On the other
hand, the $E_{accr}$ values increase from 1.53 at an equivalent size of 30 km to 1.76 at an equivalent
size of 180 km, which are 43% and 64% higher than the prescribed value (1.07). The higher
$E_{auto}$ and lower $E_{accr}$ prescribed in GCMs help to explain the issue of too frequent precipitation
events with too light precipitation intensity. The ratios of rain to cloud liquid water increase
with increasing $E_{accr}$ from 0 to 2, and then decrease after that, suggesting a possible optimal
state for the collision-coalescence process to achieve maximum efficiency for converting cloud
water into rain water at $E_{accr}$=2. The ratios of $q_r$ to $q_c$ at $E_{accr}$=1.07 and $E_{accr}$=2.0 are 0.063 and
0.142, further proving that the prescribed value of $E_{accr}$=1.07 is too small to simulate correct
precipitation intensity in models.
To further investigate what conditions and parameters can significantly influence the
enhancement factors, we classified low-level clouds according to their boundary layer
conditions and CLWPs. Both $E_{auto}$ and $E_{accr}$ increase when the boundary layer conditions
become less stable, and the values are larger in precipitating clouds (CLWP>75 gm$^{-2}$) than
those in nonprecipiting clouds (CLWP<75 gm$^{-2}$). The $E_{auto}$ values in both stable and mid-stable
boundary layer conditions are smaller than the prescribed value of 3.2, while those in unstable
boundary layers conditions are significantly larger than 3.2 regardless of whether or not the
cloud is precipitating (Table 2). All $E_{accr}$ values are greater than the prescribed value of 1.07.
Therefore, the selection of $E_{auto}$ and $E_{accr}$ values in GCMs should be regime-dependent, which
also has been suggested by Hill et al. (2015) and Walters et al. (2017).
This study, however, did not include the effect of uncertainties in GCM simulated cloud
and precipitation properties on sub-grid scale variations. For example, we did not consider the
behavior of the two enhancement factors under different aerosol regimes, a condition which
may affect precipitation formation process. The effect of aerosol-cloud-precipitation-
interactions on cloud and precipitation sub-grid variabilities may be of comparable importance
to meteorological regimes and precipitation status and deserves a further study. Other than the
large-scale dynamics, e.g., LTS in this study, upward/downward motion in sub-grid scale may
also modify cloud and precipitation development and affect the calculations of enhancement
factors. The investigation of the dependence of $E_{auto}$ and $E_{accr}$ on aerosol type and concentration
as well as on vertical velocity would be a natural extension and complement of current study.
In addition, other factors may also affect precipitation frequency and intensity even under the
same aerosol regimes and even if the clouds have similar cloud water contents. Wind shear, for
example as presented in Wu et al. (2017), is an external variable that can affect precipitation
formation. Further studies are needed to evaluate the role of the covariance of $q_c$ and $N_c$ in sub-
grid scales on $E_{auto}$ determinations, which is beyond the scope of this study and requires
independent retrieval techniques.

**Appendix A:  Joint cloud and rain LWC profile estimation**

If a time step is identified as non-precipitating, the cloud liquid water content (CLWC) profile is retrieved using Frisch et al. (1995) and Dong et al. (1998, 2014a and 2014b). The retrieved CLWC is proportional to radar reflectivity.

If a time step is identified as precipitatinging (maximum reflectivity below cloud base exceeds -37 dBZ), CLWC profile is first inferred from temperature and pressure in merged sounding by assuming adiabatic growth. Marine stratocumulus is close to adiabatic (Albrecht et a. 1990) and was used in cloud property retrievals in literature (e.g., Rémillard et al., 2013). In this study, we use the information from rain properties near cloud base to further constrain the adiabatic CLWC ($CLWC_{adiabatic}$).

Adopting the method of O'Connor et al. (2005), Wu et al. (2015) retrieved rain properties below cloud base (CB) for the same period as in this study. In Wu et al. (2015), rain drop size (median diameter, $D_0$), shape parameter ($\mu$), and normalized rain droplet number concentration ($N_W$) are retrieved for the assumed rain particle size distribution (PSD):

$$n_r(D) = N_W f(\mu) \left(\frac{D}{D_0}\right)^{\mu} \exp[-\frac{(3.67+\mu)D}{D_0}] \tag{A1}$$

To infer rain properties above cloud base, we adopt the assumption in Fielding et al. (2015) that $N_W$ increases from below CB to within the cloud. This assumption is consistent with the *in situ* measurement in Wood (2005a). Similar as Fielding et al. (2015), we use constant $N_W$ within cloud if the vertical gradient of $N_W$ is negative below CB. The $\mu$ within cloud is treated as

constant and is taken as the averaged value from four range gates below CB. Another
assumption in the retrieval is that the evaporation of rain drops is negligible from one range
gate above CB to one range gate below CB thus we assume rain drop size is the same at the
range gate below and above CB.
With the above information, we can calculate the reflectivity contributed by rain at the first
range gate above CB ($Z_r(1)$) and the cloud reflectivity ($Z_c(1)$) is then $Z_c(1) = Z(1) - Z_r(1)$,
where $Z(1)$ is WACR measured reflectivity at first range gate above CB. Using cloud droplet
number concentration ($N_c$) from Dong et al. (2014a and 2014b), CLWC at the first range gate
above CB can be calculated through
$$Z_c(1) = 2^6 \int_0^\infty n_c(r) r^6 dr = \frac{36}{\pi^2 \rho_w^2} \frac{CLWC(1)_{reflectivity}^2}{N_c} \exp(9\sigma_x^2) \tag{A2.1}$$
$$CLWC(1)_{reflectivity} = \sqrt{\frac{Z_c(1) \pi^2 \rho_w^2 N_c}{36 \exp(9\sigma_x^2)}} \tag{A2.2}$$
Where $\rho_w$ is liquid water density $n_c(r)$ is lognormal distribution of cloud PSD with
logarithmic width $\sigma_x$. Geoffroy et al. (2010) suggested that $\sigma_x$ increases with the length scale
and Witte et al. (2018) showed that $\sigma_x$ also dependent on the choice instrumentation. The
variations of $\sigma_x$ should be reflected in the retrieval by using different $\sigma_x$ values with time.
However, no aircraft measurements were available during CAP-MBL to provide $\sigma_x$ over the
Azores region. The inclusion of solving $\sigma_x$ in the retrieval adds another degree of freedom to
the equations and complicates the problem considerably. In this study, $\sigma_x$ is set to a constant
value of 0.38 from Miles et al. (2000), which is a statistical value from aircraft measurements
of marine low-level clouds.
We then compare the $CLWC_{adiabatic}$ and the one calculated from $CLWC_{reflectivity}$ at the
first range gate above CB. A scale parameter (*s*) is defined as $s = \frac{CLWC_{reflectivity}(1)}{CLWC_{adiabatic}(1)}$ and the
entire profile of $CLWC_{adiabatic}$ is multiplied by *s* to correct the bias from cloud sub-
adiabaticity. Reflectivity profile from cloud is then calculated from Eq. (A2.1) using the
updated $CLWC_{adiabatic}$ and the remaining reflectivity profile from WACR observation is
regarded as rain contribution. Rain particle size can then be calculated given that $N_W$ and $\mu$ are
known and rain liquid water content (RLWC) can be estimated.
There are two constrains used in the retrieval. One is that the summation of cloud and rain
liquid water path (CLWP and RLWP) must be equal to the LWP from microwave radiometer
observation. Another is that rain drop size ($D_0$) near cloud top myst be equal or greater than 50
$\mu m$ and if $D_0$ is less than 50 $\mu m$, we decrease $N_W$ for the entire rain profile within cloud and
repeat the calculation until the 50 $\mu m$ criteria is satisfied.
It is difficult to quantitatively estimate the retrieval uncertainties without aircraft in situ
measurements. For the proposed retrieval method, 18% should be used as uncertainty for
RLWC from rain properties in Wu et al. (2015) and 30% for CLWC from cloud properties in
Dong et al. (2014a and 2014b). The actual uncertainty depends on the accuracy of merged
sounding data, the detectability of WACR near cloud base and the effect of entrainment on
cloud adiabaticity during precipitating. In the recent aircraft field campaign, the Aerosol and
Cloud Experiments in Eastern North Atlantic (ACE-ENA) was conducted during 2017-2018
with a total of 39 flights over the Azores, near the ARM ENA site on Graciosa Island. These
aircraft in situ measurements will be used to validate the ground-based retrievals and
quantitatively estimate their uncertainties in the future.
Figure A1 shows an example of the retrieval results. The merged sounding, ceilometer,
microwave radiometer, WACR and ceilometer are used in the retrieval. Whenever one or more
instruments are not reliable, that time step is skipped, and this results in the gaps in the CLWC
and RLWC as shown in Figures A1(b) and A1(c). When the cloud is classified as
nonprecipitating, no RLWC will be retrieved as well. Using air density ($\rho_{air}$) profiles
calculated from temperature and pressure in merged sounding, mixing ratio ($q$) can be
calculated from LWC using $q(z) = LWC(z)/\rho_{air}(z)$.
**Acknowledgements**
The ground-based measurements were obtained from the Atmospheric Radiation Measurement
(ARM) Program sponsored by the U.S. Department of Energy (DOE) Office of Energy
Research, Office of Health and Environmental Research, and Environmental Sciences
Division. The data can be downloaded from http://www.archive.arm.gov/. This research was
supported by the DOE CESM project under grant DE-SC0014641 at the University of Arizona

through subaward from University of Maryland at Baltimore County, and the NSF project under grant AGS-1700728 at University of Arizona. The authors thank Dr. Yangang Liu at Brookhaven National Laboratory for insightful comments and Ms. Casey E. Oswant at the University of Arizona for proof reading the manuscript. The three anonymous reviewers are acknowledged for constructive comments and suggestions which helped to improve the manuscript.

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

**Table 1. The parameters of autoconversion and accretion formulations for four**
**parameterizations.**

| | $A$ | $a1$ | $a2$ | $B$ | $b$ |
|---|---|---|---|---|---|
| Khairoutdinov and Kogan (2000) | 1350 | 2.47 | -1.79 | 67 | 1.15 |
| Liu and Daum (2004) | $1.3 \times 10\beta_6^6,$ where $\beta_6^6 = [(r_v + 3)/r_v]^2,$ $r_v$ is mean volume radius. modification was made by Wood (2005b) | 3 | -1 | N/A | N/A |
| Tripoli and Cotton (1980) | 3268 | 7/3 | -1/3 | 1 | 1 |
| Beheng (1994) | $3 \times 10^{34}$ for $N_c < 200$ cm$^{-3}$ 9.9 for $N_c > 200$ cm$^{-3}$ | 4.7 | -3.3 | 1 | 1 |


**Table 2. Autoconversion (left) and accretion (right) enhancement factors in different**
**boundary layer conditions (LTS > 18 K for stable, LTS < 13.5 K for unstable and LTS**
**within 13.5 and 18 K for mid-stable) and in different LWP regimes (LWP ≤ 75 g m$^{-2}$ for**
**non-precipitating and LWP > 75 g m$^{-2}$ for precipitating).**

|  | LWP ≤ 75 g m$^{-2}$ | LWP > 75 g m$^{-2}$ |
|---|---|---|
| LTS > 18 K | 2.32/1.42 | 2.75/1.52 |
| 13.5 ≤ LTS ≤ 18K | 2.61/1.47 | 3.07/1.68 |
| LTS < 13.5 K | 4.62/1.72 | 6.94/1.86 |


**Table 3. Autoconversion and accretion enhancement factors ($E_{auto}$ and $E_{accr}$) for the**
**parameterizations in Table 1 except the Khairoutdinov and Kogan (2000) scheme. The**
**values are averaged for 60-km and 180-km equivalent sizes.**

| | $E_{auto}$ | | $E_{accr}$ | |
|---|---|---|---|---|
| | 60-km | 180-km | 60-km | 180-km |
| Liu and Daum (2004) | 3.82 | 4.23 | N/A | N/A |
| Tripoli and Cotton (1980) | 2.46 | 2.69 | 1.47 | 1.56 |
| Beheng (1994) | 6.94 | 5.88 | 1.47 | 1.56 |


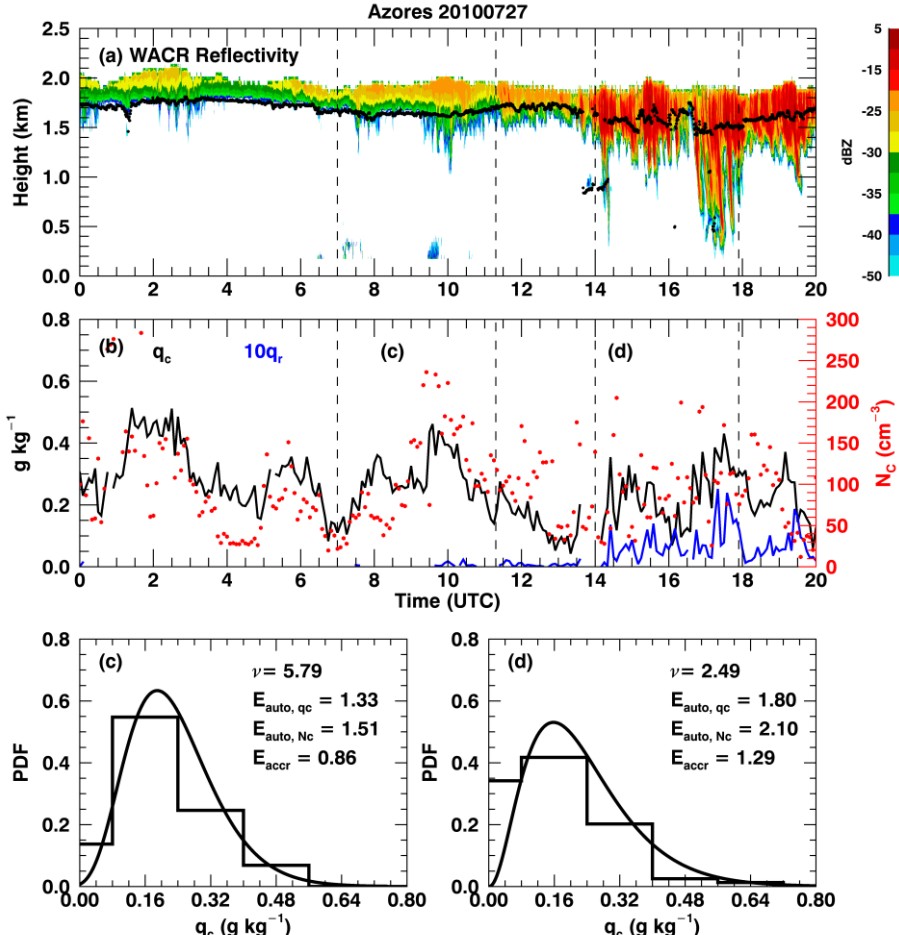


**Figure 1. Observations and retrievals over Azores on 27 July 2010. (a) W-band ARM cloud radar (WACR) reflectivity (contour) superimposed with cloud-base height (black dots). (b) Black line represents averaged cloud water mixing ratio ($q_c$) within the top five range gates, blue line represents averaged rain (×10) water mixing ratio within five range gates around maximum reflectivity, red dots are the retrieved cloud droplet number concentration ($N_c$). Dashed lines represent two periods that have 60 km equivalent sizes with similar $\overline{q_c}$ but different distributions as shown by step lines in (c) and (d). Curved lines in (c) and (d) are fitted gamma distributions with the corresponding shape parameter ($\nu$) shown on the upper right. $N_c$ distributions are not shown. The calculated autoconversion ($E_{auto,\,qc}$ from $q_c$ and $E_{auto,\,Nc}$ from $N_c$) and accretion ($E_{accr}$) enhancement factors are also shown.**

847

848

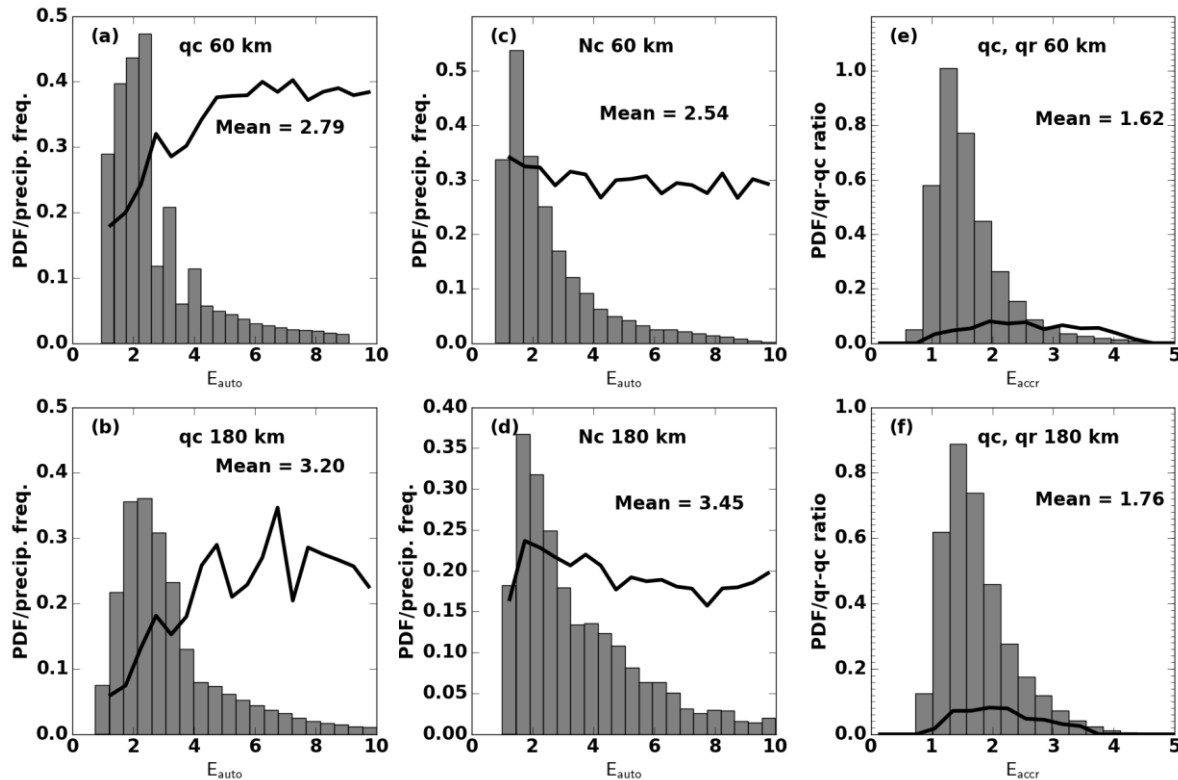

849

**Figure 2.** Probability density functions (PDFs) of autoconversion (a - d) and accretion (e - f) enhancement factors calculated from $q_c$ (a-b), $N_c$ (c-d), and the covariance of $q_c$ and $q_r$ (e-f). The two rows show the results from 60-km and 180-km equivalent sizes, respectively, with their average values. Black lines represent precipitation frequency in each bin in (a)-(d) and the ratio of layer-mean $q_r$ to $q_c$ in (e)-(f).

855

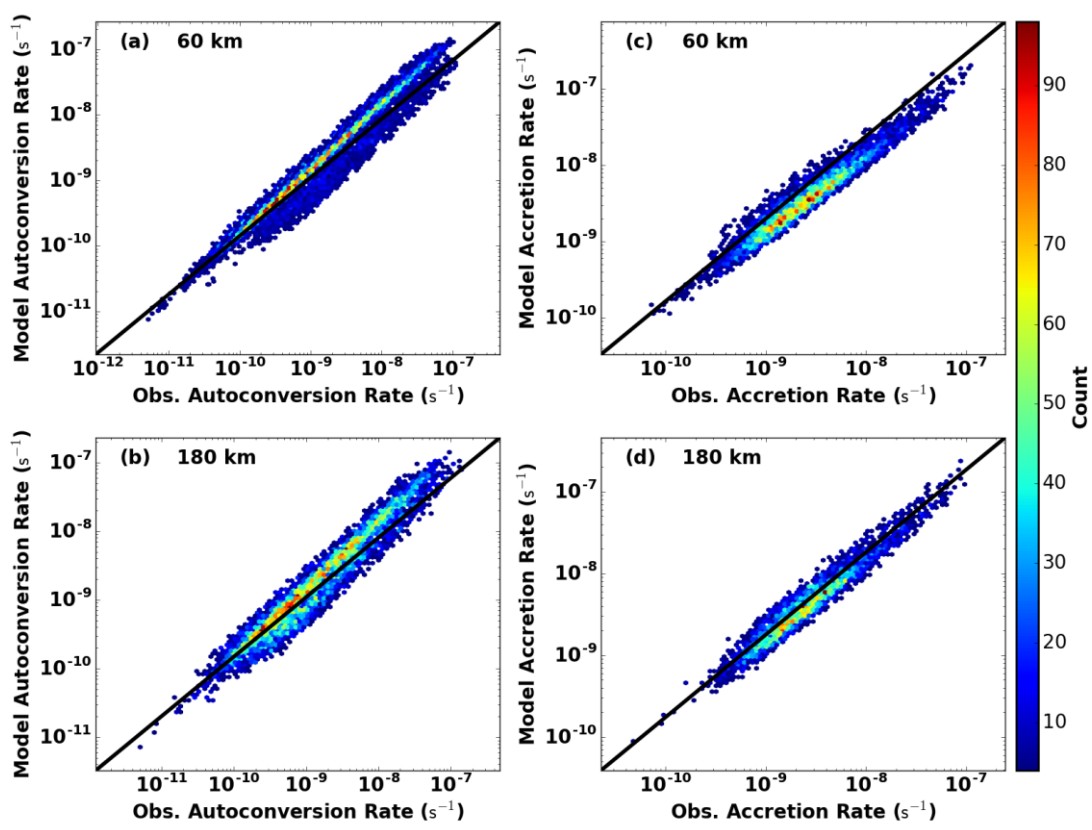

856

**Figure 3. Comparison of autoconversion (a-b) and accretion (c-d) rates derived from observations (x-axis) and from model (y-axis). Results are for 60-km (a and c) and 180-km model equivalent sizes. Colored dots represent joint number densities.**

860

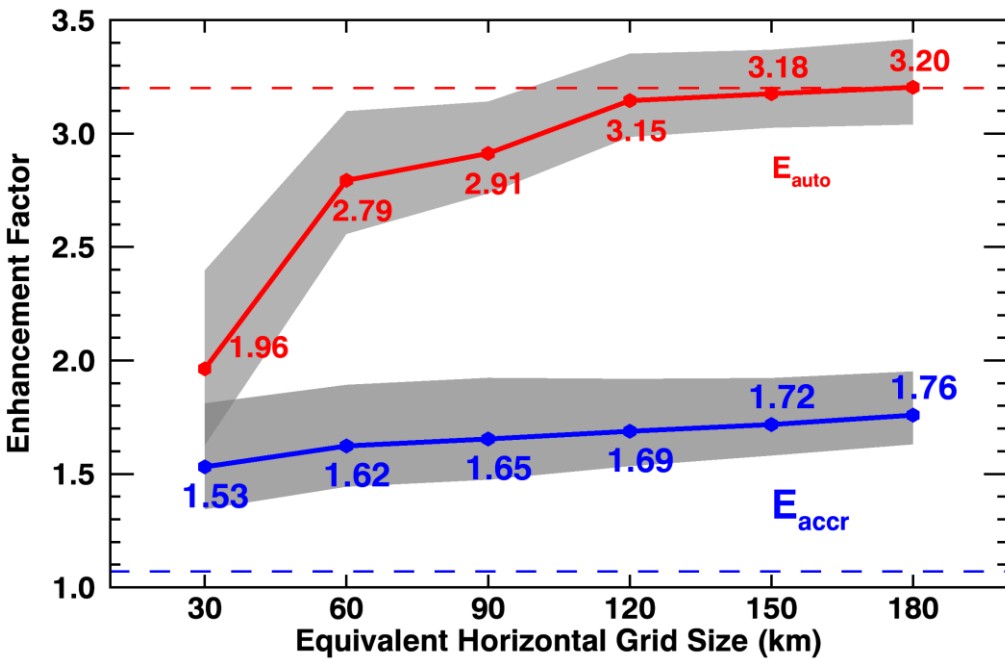

Figure 4. Autoconversion (red line) and accretion (blue line) enhancement factors as a function of equivalent sizes. The shaded areas are calculated by varying $q_c$ and $q_r$ within their retrieval uncertainties. The two dashed lines show the constant values of autoconversion (3.2) and accretion (1.07) enhancement factors prescribed in MG08.

865

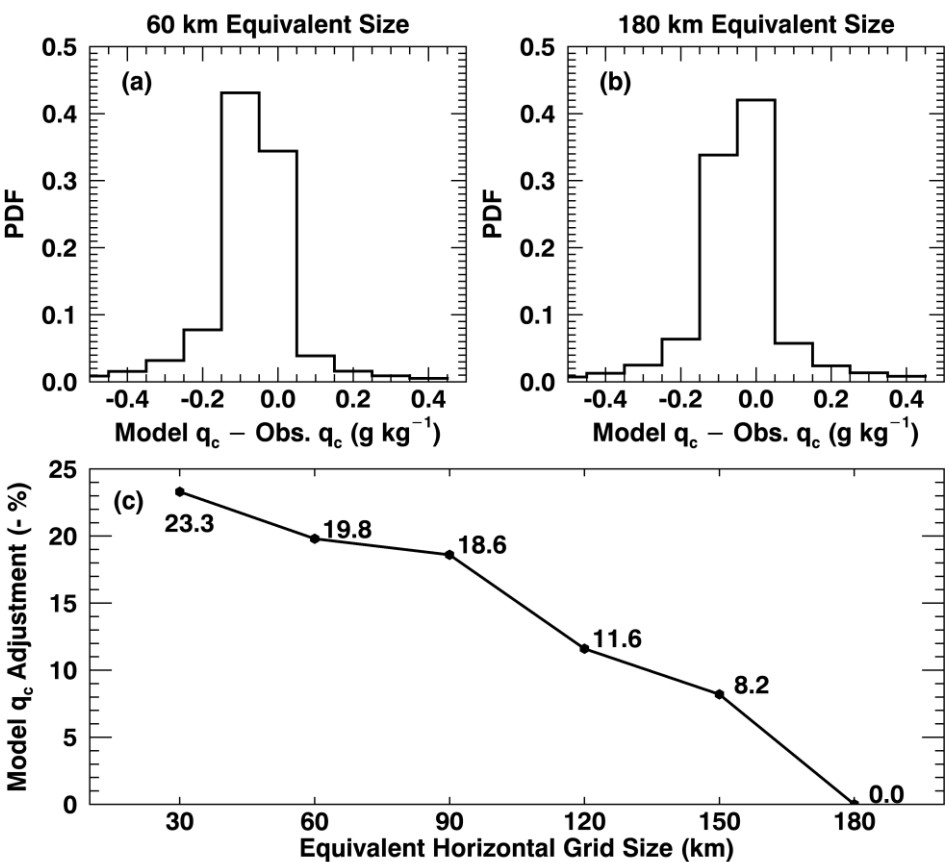

866

Figure 5. $q_c$ needed for models to adjust to reach the same autoconversion rate as observations for (a) 60-km and (b) 180-km model equivalent sizes. Positive biases represent increased $q_c$ are required in models and negative biases mean decreased $q_c$. The average percentages of adjustments for different equivalent sizes are shown in panel (c) and note that the percentages in the vertical axis are negative.

872

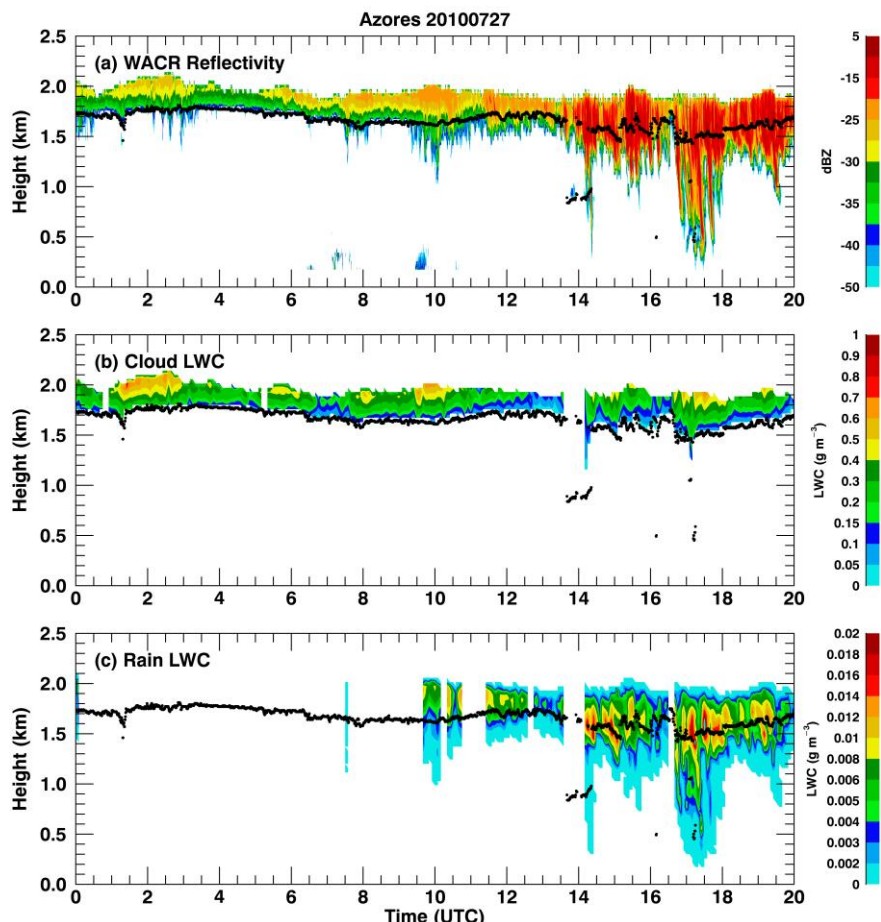

873

**Figure A1. Joint retrieval of cloud and rain liquid water content (CLWC and RLWC) for the same case as in Figure 1. (a) WACR reflectivity, (b) CLWC, and (c) RLWC. The black dots represent cloud base height. Blank gaps are due to the data from one or more observations are not available or reliable. For example, the gap before 14 UTC is due to multiple cloud layers are detected whereas we only focus on single layer cloud.**