# Peer review of "1Evaluation of autoconversion and accretion enhancement factors in GCM warm-rain 2parameterizations using ground-based measurements at the Azores"

_Atmospheric Chemistry and Physics, 2018_

## Referee Comment (RC1) · Anonymous Referee #1 · 17 Jul 2018

The goal of this study is to extend the results of studies such as Lebsock et al. (2013) and Boutle et al. (2014) on quantifying the effects of sub-grid scale inhomogeneity on microphysical process rates applied in GCMs from observations. The central tenet is that inhomogeneity varies with length scale and meteorological regime, thus the currently standard use of "universal" constants to characterize inhomogeneity cannot adequately describe subgrid-scale variability across a range of horizontal grid sizes or environmental conditions. The authors use a temporally extensive remote sensing dataset primarily sampling shallow convection over Graciosa Island in the Azores to develop "scale-aware" enhancement factors for the autoconversion and accretion processes ($E_{auto}$ and $E_{accr}$, respectively) for several commonly used bulk microphysical

parameterizations. These enhancement factors are estimated from compositing of variances and covariances of instantaneous retrievals of cloud and rain liquid water path (CLWP and RLWP, respectively) and cloud drop number concentration $N_c$ over varying time windows, which the authors argue are roughly equivalent to a GCM horizontal grid length if a constant wind speed is assumed.

I agree with the authors' basic premise that the use of constant values for $E_{auto}$ and $E_{accr}$ in GCM microphysics schemes is unrealistic and likely introduces precipitation biases similar (perhaps in magnitude if not sign) to assuming that grid-mean quantities (e.g. of $N_c$ and cloud and rain liquid water mixing ratios $q_c$ and $q_r$) are applicable to calculation of process rates in models with coarse grids (say horizontal grid length $L$ greater than a kilometer or so). Furthermore, their assertion that enhancement factors should vary as a function of $L$ as well as meteorological regime is well-stated, although they are not able to access independent information on aerosol-cloud interactions, which I suspect may be of comparable importance to the stability and LWP criteria analyzed.

Despite agreeing with the importance and timeliness of the premise of the manuscript, I have several major issues with the relevance of the observations to diagnosis of microphysical process inhomogeneity. Most importantly, the retrievals of cloud and rain/drizzle properties are not collocated; drizzle properties are only retrieved below cloud base. Cloud and drizzle properties are convolved within cloud such that what is classified as CLWP in fact includes contributions from in-cloud drizzle as well. Microphysical process rate equations assume coincident cloud and rain water mixing ratios (accretion) and coincident cloud water and drop number concentration (autoconversion), so unless it could be shown from some other dataset (LES? Aircraft observations? Maybe even a simplified 1D model?) that subcloud RLWP correlates highly with in-cloud RLWP and has similar magnitude, I have serious doubts about the physical relevance of the retrieved covariances. This may explain the apparently low ratios of cloud to rain water presented in the paper (see lines 33-34 and 291-293, Fig. 2e-f),

although the authors give no "expected" value of this ratio for comparison.

The use of column-integrated liquid water paths introduces further uncertainty because the partitioning of the collision-coalescence process into autoconversion and accretion sub-processes is heterogeneous in the vertical. In the shallow clouds typical of the ENA site, autoconversion will be dominant near cloud top where cloud droplets have reached a maximum size due to condensation and larger drizzle drops are rare while accretion dominates lower in cloud, where the drizzle drops initially formed at cloud top sediment and continue to grow by collecting cloud droplets. Erasing this coherent vertical variability by the use of integrated water paths may bias the results presented: in stratiform clouds, liquid water is at a maximum near cloud top (i.e. CLWP is weighted toward cloud top), such that the $E_{accr}$ values in particular are using over-inflated liquid water values. I'm also confused about how the authors transformed liquid water paths to mixing ratios. They state that "CLWC [cloud liquid water content] values are transformed to $q_c$...by dividing by air density" (lines 191-192) and similar for $q_r$ (lines 194-195) but never define how they calculate CLWC or drizzle LWC. Are they dividing water path by cloud/drizzle shaft depth for an average value? Or are they applying the methods of Xie and Zhang (2015) and Wu et al. (2015) to the retrievals? Is the retrieval of $N_c$ vertically resolved? This part of the methodology is insufficiently described to understand what the authors did, and regardless, it doesn't address the issue that drizzle properties can only be retrieved below cloud using their approach.

Finally, the authors made no attempt to quantify the uncertainty of the reported enhancement factors, such that I cannot make a determination as to whether their $E_{auto}$ and $E_{accr}$ are statistically distinct from the constant values introduced by Morrison and Gettelman (2008). This is particularly relevant to Figure 4. I would also have liked to see the authors show the quantitative impact of treating $q_c$ and $N_c$ individually with respect to calculating $E_{auto}$, as their derivation of Equation 4 assumes that the covariability of $q_c$ and $N_c$ can be ignored. While the magnitude of $E_{auto}$ is comparable for $q_c$ or $N_c$ individually, I don't have a good sense for what including variability of both

[Figure]

variables implies for the predicted $E_{auto}$ values. It's certainly a problem that CLWP and $N_c$ are correlated in the ARM dataset employed, but that doesn't change the fact that variability of $N_c$ is likely substantial, especially for the longer time periods analyzed or in more cumuliform precipitation.

In light of these concerns, I must recommend that this manuscript be **rejected** in its current form. A revised version of the manuscript only addressing autoconversion would be more feasible and would also be very useful to the parameterization development community, although as mentioned above, I would ask that the authors address the question of whether ignoring covariability of $q_c$ and $N_c$ is a reasonable assumption. I would be happy to review a revised and refocused manuscript.

Until remote sensing datasets can unambiguously partition in-cloud condensed water into cloud and drizzle components, analysis of cloud-rain covariance from the present spatially disjoint cloud and rain retrievals cannot be used to inform accretion parameterizations. A technique like that of Luke and Kollias (2013; doi:10.1175/JTECH-D-11-00195.1) that uses skewness of the Doppler spectrum to differentiate between cloud and drizzle could be combined with a method similar to Frisch et al. (1998; doi:10.1029/98JD01827) to retrieve vertically-resolved profiles of cloud and rain water, albeit likely only in stratiform clouds. If such an approach could be developed, the analysis performed in this manuscript would be more tractable although it would likely need to be validated before application to the GCM cloud inhomogeneity problem given the amount of technical work necessary to provide confidence in the retrievals.

---

## Referee Comment (RC2) · Anonymous Referee #2 · 18 Jul 2018

This paper discusses how variability of cloud and rain at the GCM sub-grid scale affect the parametrizations of autoconversion and accretion that are typically used. This has become a popular topic in recent years with many papers and modelling centres using this as a method of improving warm rain simulation. The current paper has some novel aspects, for example the use of data from the Azores to evaluate parametrizations, but I feel would require some significant modifications before it is acceptable for publication.

Major comments:

1. I don't feel this paper fully or correctly acknowledges the previous work that has been done in this field, which leads to many statements with are either misleading,

incorrect, in contradiction to previous studies without explanation, or presented as new when actually they have been published before. Specific examples of this are:

a) L31, 284, 390 and elsewhere - repeatedly the authors refer to "GCMs", implying that they are stating a common feature of many models, whereas in actual fact they are referring specifically to the MG08 microphysics scheme which is only used in a very small number of GCMs. This terminology needs to be more precise, to highlight the fact that not all GCMs make the same assumptions as MG08.

b) L99 - this statement is incorrect - whilst some models do use prescribed values regardless of meteorological conditions, the whole point of Boutle et al (2014), which is cited as introduction to this statement, is to provide a parametrization depending on meteorological conditions which can be used in GCMs. This parametrization is improved upon by Hill et al (2015), who add in a regime dependence to the parametrization, and implemented in a model by Walters et al (2017). The authors need to acknowledge this work in the context of their own.

c) L293-294 - this statement is just repeating the previous conclusions of Boutle et al (2014) and Lebsock et al (2013).

d) L335 - Hill et al (2015) also show regime dependence and should be cited here.

e) L336-337 - I don't understand this statement - why is it difficult to vary enhancement factors in GCMs? Walters et al (2017) using the parametrizations of Boutle et al (2014) does exactly that - there is nothing difficult here and no reason why other GCMs could not do similar.

f) L364-368 - I don't fully understand what is being claimed here, and it certainly is not supported by any evidence presented in the paper. But what I think the authors are saying is that in more cumulus-type (less stratiform) clouds, E_auto should be smaller. This appears contradictory to the results of Boutle et al (2014) (their Fig 10) and Hill et al (2015) which show that E_auto is higher in convective type cloud regimes. It also

appears in contradiction to the authors own statement on L429-430 (a statement that appears with no justification or background), that unstable boundary layers give rise to larger E_auto values. Please clarify this.

g) L433 - as is done in Hill et al (2015) and Walters et al (2017).

h) Fig 4 - despite the constant criticism of MG08 for using a fixed value of E_auto=3.2, this figure shows that at larger grid sizes, this value is actually incredibly good - some credit should be given to MG08 for this!

2) L148, L151 - equations 4 and 5 are incorrect, the term in the denominator should be Gamma(nu) not Gamma(a) as written (see Eq 7 of Boutle et al (2014) or Eq 6 of Pincus and Klein (2000)). I hope this is only a typo and not a problem with all of the data analysis! Also, I'm confused about whether or not you are investigating variability of Nc - the text seems to suggest you are, but this equation ignores any variability in Nc - please clarify the text and correct the equation if necessary.

3) L207, 340 - simply using a constant wind speed is quite crude - most previous studies with ground based equipment (eg. Boutle et al 2014) have either used actual wind speeds or model derived reanalysis wind speeds to construct spatial scales from time averages. At the very least this simplification needs to be noted and possible errors due to this discussed.

4) L220 onwards, L281, elsewhere - the analysis appears to be presented in terms of LWP and RWP, i.e. column integrals of quantities. This is very different to the LWC and RWC, i.e. grid-box mean quantities which are used in parametrizations. Most previous studies have used LWC and RWC to calculate the variability, and so the results are directly applicable to parametrizations. It's not clear to me that results presented in LWP and RWP are so directly applicable. The authors need to investigate how applicable their results using column-integral quantities are to previous studies and parametrizations - it appears from the text that you do have direct observations of LWC and RWC, so it should not be too difficult to make this comparison, or re-do the

analysis using the LWC and RWC data.

General comments:

Title - should probably be "Evaluation of ..."

L50 - should say "a significant amount of drizzle is evaporated"

L56 - I'm not entirely sure I agree with this statement - change in albedo (i.e. the first indirect effect) is the most significant indirect effect. There is also an extensive literature on buffering of the 2nd indirect effect and mechanisms through which aerosol could even enhance convective precipitation. At the very least this statement needs to be more accurate in the context it is being used - increases in aerosol are mainly thought to suppress precipitation in MBL clouds.

L62 - MG08 is an odd reference here, given it discusses a microphysics parametrization, something which is required in models of all scales

L63 - the "process" of autoconversion and accretion only exist because modellers have partitioned the liquid water into "cloud" and "rain" categories - please rephrase this sentence, they are not real processes, all that happens in the real atmosphere is collision-coalescence of water droplets.

L64, 72, 73, 122, 129 - the references to MG08 and LG13 are odd here, given they do not propose autoconversion or accretion parametrizations of their own, they use the scheme of KK00 which is already referenced.

L77 - using a prime to denote grid-mean quantities is somewhat non-standard - an overbar is the more typical symbol for a mean quantity.

L79 - I'm not sure I follow why positive skewness is important - can you elaborate? It is only really the non-linear form of the equations that mean rates depend strongly on the sub-grid variability.

L100 - Boutle et al (2014) use a combination of aircraft, ground-based and satellite

Interactive
comment

measurements.

L312 - using flash flooding as an example when discussing drizzling marine stratocumulus is a bit of a leap, I suggest removing this statement unless you have any evidence that extreme rainfall rates are affected.

References:

Boutle, I. A., Abel, S. J., Hill, P. G. and Morcrette, C. J. (2014). Spatial variability of liquid cloud and rain: observations and microphysical effects. Q. J. R. Meteorol. Soc., 140, 583-594, doi:10.1002/qj.2140

Hill, P. G., Morcrette, C. J. and Boutle, I. A. (2015). A regime-dependent parametrization of subgrid-scale cloud water content variability. Q. J. R. Meteorol. Soc., 141, 1975-1986, doi:10.1002/qj.2506

Walters, D., Baran, A., Boutle, I., Brooks, M., Earnshaw, P., Edwards, J., Furtado, K., Hill, P., Lock, A., Manners, J., Morcrette, C., Mulcahy, J., Sanchez, C., Smith, C., Stratton, R., Tennant, W., Tomassini, L., van Weverberg, K., Vosper, S., Willett, M., Browse, J., Bushell, A., Dalvi, M., Essery, R., Gedney, N., Hardiman, S., Johnson, B., Johnson, C., Jones, A., Mann, G., Milton, S., Rumbold, H., Sellar, A., Ujiie, M., Whitall, M., Williams, K. and Zerroukat, M. (2017). The Met Office Unified Model Global Atmosphere 7.0/7.1 and JULES Global Land 7.0 configurations. Geosci. Model Dev., doi:10.5194/gmd-2017-291

Pincus, R., and S. A. Klein (2000), Unresolved spatial variability and microphysical process rates in large‐scale models, J. Geophys. Res., 105(D22), 27059–27065, doi: 10.1029/2000JD900504

Lebsock, M., H. Morrison, and A. Gettelman (2013), Microphysical implications of cloud‐precipitation covariance derived from satellite remote sensing, J. Geophys. Res. Atmos., 118, 6521–6533, doi: 10.1002/jgrd.50347

---

## Referee Comment (RC3) · Anonymous Referee #3 · 20 Aug 2018

ACPD Review of "Evaluate autoconversion and accretion enhancement factors in GCM warm-rain parameterizations using ground-based measurements at the Azores" Manuscript ID: ACP-2018-499 Authors: P. Wu, B. Xi, X. Dong, and Z. Zhang

This manuscript uses ground-based observations and retrievals from the DOE ARM Mobile Facility at Graciosa Island over the Azores to calculate and characterize shallow cloud autoconversion and accretion enhancement factors as a function of temporal and spatial size, which help establish the observational benchmarks with which to compare against autoconversion and accretion factors in GCM parameterizations. This is a worthy goal, as many GCMs precipitate both too frequently and too lightly relative

to observations, with the preliminary inference that the sub-grid enhancement factor for autoconversion in many parameterizations is too strong, and the factor for accretion is too weak. This is because autoconversion primarily determines precipitation initiation, and accretion primarily controls drizzle/precipitation intensity. The approach taken in this study both interesting and instructive, as the homogeneity of cloud/liquid water path (LWP) properties tends to be higher for smaller grid sizes, suggesting that GCMs with finer resolutions many have an even weaker accretion enhancement factor than coarser models. This appears to be an important consideration for models with a range of grid resolutions. While autoconversion and accretion biases are systematically characterized in this study, the additional novelty of this work is that the biases are not fixed, but rather regime dependent, with both lower tropospheric stability and precipitation playing a significant role. The layout and results of this study have the potential to inform existing and future parameterizations about how to tailor precipitation enhancement factors as a function of local thermodynamics, resolution, temporal length, and precipitation itself. While other regions will need to be studied as well to gather and calculate additional autoconversion and accretion enhancement factors in other regimes and/or large-scale dynamics, this study is a good start in providing potential guidance for the modeling and model analysis communities. While there is much potential in this manuscript, there are also a number of minor to moderate technical and science questions which need addressing prior to consideration for publication, and these are mentioned below. Though the manuscript was not intended to be an exhaustive study of all the factors that that may modulate the autoconversion and accretion enhancement factors, it might be complementary to the study to also consider a few additional large-scale factors, such as local vertical velocity profiles, as has recently been done in work examining entrainment velocities over the MAGIC campaign over the Northeast Pacific, in which even during the boreal summer, approximately 20% of the profiles were observed to have rising motion near cloud top. Whether rising motion would enhance accretion rates/enhancement factors might complement the findings presented in this study. Finally, at the end of this review is an enumeration of grammat-

ical suggestions/typos; the list is non-exhaustive such that a thorough proofreading will be essential prior to publication.

Science/Technical Questions and Comments: 1) On lines 33-34 (and also lines 291-293), the authors state that "the ratios of rain to cloud liquid water at Eaccr=1.07 and Eaccr=2.0 are 0.048 and 0.119, respectively, further proving that the prescribed value of Eaccr=1.07 used in GCMs is too small to simulate precipitation intensity", but it is somewhat unclear to me how this proves this, unless the authors include (for clarity) the respective GCM RLWP-to-CLWP ratios as well in this statement. How does this range of ratios compare to those from Lebsock et al. (2011)? Based on their Figure 6, it seems that the RLWP-to-CLWP ratio is generally higher than 0.119, though of course other factors are work as well (e.g. cloud top effective radius, included in their Figure 8). If I am interpreting these differences correctly, what do the authors attribute to the apparently higher ratios in that observational study? Reference: Lebsock, M. D., T. S. L'Ecuyer, and G. L. Stephens, 2011: Detecting the ratio of rain and cloud water in low-latitude shallow marine clouds. J. App. Meteor. Climatology, 50, 419-432, doi: 10.1175/2010JAMC2494.1. 2) Lines 77-78 (and elsewhere): The authors use primes (') to denote grid means, and are consistent about this, but wouldn't an overbar typically denote grid mean values? Often, but not always, primes are designated for deviations from the mean. Overall, this is a fairly minor comment. 3) Lines 144-145, and more generally the implications for the findings of this study: Is the shape parameter, $\gamma$, somewhat analogous to the LWP homogeneity from Wood and Hartmann 2006 (which was simply the squared quantify of the ratio of the mean LWP to its standard deviation)? In that study, precipitation was not explicitly examined, but instead the brokenness of cloud fields was found to increase (total CF decrease) with decreasing LWP homogeneity. The greater shallow convective cellular structure with decreasing LWP homogeneity, however, may have been more conducive for heavier precipitation.

In this study, the relationship appears to be the link between reduced LWP homogeneity or other cloud field homogeneity and an increase in precipitation intensity. The latter is

often too small in climate models, and too homogeneous of fields may be the culprit. Perhaps even though the authors have cited Barker et al. (1996); Pincus et al. (1999), and Wood and Hartmann (2006), an even stronger parallel/analogy should be made between the objectives and findings here and those from those studies, particularly in the discussion section later on as appropriate. 3) Results in Figure 1: Traditionally, we think of the large number concentrations decreasing with precipitation, as heavier precipitation is dominated by fewer, larger drops. It perhaps seems a little surprising that a peak (red bin denoting Nc) emerges during the heavy drizzle phase of Nc values of 100 cmˆ-3 (Figure 1d). However, there are fewer very high values of Nc (e.g. >150 cmˆ-3 ) during the hours of moderate to heavy drizzle. From Fig. 1b as well, there does seem to be high values of Nc just before or after periods of heavier drizzle, but a suppression of Nc is observed somewhat during the stronger pulses of precipitation. Can the authors discuss perhaps why more of the larger number concentrations are not removed during the heavier drizzle events? On the other side of the spectrum, during most of the non-precipitating phase, there is a local maximum of Nc at 150 cmˆ-3. Have the authors considered looking at the corresponding time series of effective radius to include as perhaps another panel for Figure 1? This may complement the Nc observations quite nicely.

4) Lines 196-197: Is this the traditional LTS definition of potential temperature at 700 hPa minus potential temperature at near the surface (or 1000 hPa)?

5) Figure 2: Generally, the precipitation frequency ranges from about ∼0.1 to just above 0.4, which seems a little low (at least the highest end of precipitation incidence) compared to other observational studies (e.g. Kubar et al. 2009). What is the definition of precipitation frequency here – is it any precipitation observed in the column, or only that which reaches the surface? Reference: Kubar, T. L., D. L. Hartmann, and R. Wood, 2009: Understanding the importance of microphysics and macrophysics in marine low clouds, Part I: satellite observations. J. Atmos. Sci., 66, 2953-2972, doi: 10.1175/2009JAS3071.1 6) Is the E_auto critical threshold of 4 of converting cloud to

drizzle drops found by the authors from the results in Figure 2 considered a novel finding, or has this been reported elsewhere as well (or perhaps a slightly different threshold)? 7) Lines 284-285 and Figure 4: The authors cite and use 1.07 as a representative value for E_accr (based on Morrison and Gettleman 2008); is this a fairly common value used in other GCM parameterizations as well? Presumably, based on this study, the range for E_accr in climate models is smaller than the observed/calculated 1-4 range for E_accr found in this study. 8) Lines 354-354 and implications of study: As alluded to in the Introductory remarks of this review above, an interesting finding of this study is that the enhancement factors are even more different/biased in finer-resolution GCMs versus observations than coarser-resolution models. Thus, even though a frequent goal is improving resolution in simulations, more care is needed to address the "too frequent/too light" precipitation problem. This study appears to be instructive in how to potentially overcome this barrier. 9) Also, as alluded to in the introduction, while the autoconversion/accretion enhancement factor dependence on both scale and LTS regime is very intriguing, the authors may also want to expand (or propose for future work) the dependence on either near cloud-top vertical velocity (e.g. from reanalysis data such as ECMWF) and/or boundary layer vertical velocity. My assumption might be that the behavior may be similar to stability; for upward motion near cloud top, both autoconversion/accretion factors may be higher (as they are for reduced LTS), but it would be interesting to know how comparable such an effect may be to LTS. In a somewhat similar vein, I do commend the authors for discussing that other variables (e.g. aerosol type and concentration) may be important as well for the two enhancement factors studied in this investigation in the very last paragraph. This at least sets up where the authors or others can proceed to continue to expand this line of research. 10) Lines 398-399: The authors list a number of studies from the ∼mid-2000s which discuss existing parameterizations, but are the latest GCMs quite similar as well? Is there a good recent paper or series of papers which discuss recent parameterization updates, if they exist?

Minor Notes and Grammatical Suggestions/Typos: (*Note – this is a thorough, albeit still incomplete list of typos. Please professionally edit this manuscript prior to resubmitting.) 1) Line 28: change "increase" to "increases" 2) Line 50: change "drizzle are" to "drizzle is" 3) Line 62: change "Example" to "example" 4) Line 65: change "process that drizzle drops" to "process of drizzle drops" 5) Line 212: change "19-month" to "19 months" 6) Line 221: add an "a" before "more homogeneous" 7) Line 228: add an "a" prior to "similar" 8) Line 231: change "contribute" to "contributes" for proper subject ("combination")-verb ("contributes") agreement 9) Line 240: add "the" before "cloud" 10) Line 246: change "5-hour" to "5-hours"; similarly, for line 251: change "2-hour" to "2-hours" 11) Line 259: add "the" prior to "autoconversion" 12) Line 273: add "a" before "similar" 13) Line 317: Consider changing "seem easier to produce drizzle" to "more easily produce drizzle" 14) Line 372: change "are representing" to "represents". Also, in general the sentence from Line 371 – 373 is slightly awkward and probably should be rewritten. 15) Line 378: change "associate" to "associated" 16) Line 385: add "a" before "variety" 17) Line 659: add "are for" before "2-hr"

---

## Author Response (AR1)

**A Point-by-point Response to Review Comments**

Dear Dr. Feingold,

We are submitting the revised manuscript (#acp-2018-499) for your consideration of publication in *Atmospheric Chemistry and Physics*. We have carefully studied the reviewers' comments and revised the manuscript accordingly. Please find the point-by-point response (marked as blue) to the review comments. We have provided a copy of track-change manuscript as well as a clean copy of the revised manuscript.

Please note that we submitted the response to reviewer #1 and #2 before the comment from reviewer #3 was posted. In this submission, we made revisions according to all reviewers' comments and made changes to the line numbers in the response to reviewer #1 and #2 accordingly.

Thank you for your consideration of this submission. We hope you find our response adequately address the review comments and the revision acceptable. We would greatly appreciate it if you could get back to us with your decision at your earliest convenience.

Sincerely,

Peng Wu

Department of Hydrology and Atmospheric Sciences

University of Arizona

Tucson, AZ 85721, USA

**General response to all reviewers**
We would like to thank you for the constructive comments and suggestions. We appreciate your time. As you will see, the manuscript has been revised follow each reviewer's suggestions. In the response, black are reviewers' comments and blue are our responses.

The major changes are:

1.  The cloud and rain water mixing ratios are now collocated, and the method is described in Appendix A in the revised manuscript. We combine remote sensing and adiabatic assumption to jointly estimate cloud and rain liquid water content (CLWC and RLWC) within the cloud layer. We also estimate the uncertainties in enhancement factor based on our retrieval uncertainties and the results are shown in the updated Figure 4.

2.  Following the suggestion of Reviewer 1, $E_{auto}$ and $E_{accr}$ in the revised manuscript are calculated at different layers of cloud to reveal the physical processes. We use averaged $q_c$ in top five range gates to calculate $E_{auto}$ and averaged $q_c$ and $q_r$ in five range gates around maximum reflectivity to calculate $E_{accr}$. Despite substantial changes in the data used in calculations, the trend of the new results is similar to previous one except that the values slightly increase. Thus, most of our conclusions still hold.

3.  Instead of roughly assuming 10 m s$^{-1}$ horizontal wind, we now use the mean wind speed within a cloud layer from ARM merged sounding data. The terminology is changed from '2-hour…5-hour time intervals' to '60-km and 180-km model grids' as we mimic the specific model grid sizes instead of specific time intervals.

4.  Suggested by Reviewer 2, we did extensive literature reviews and rephrased sentences in both introduction and discussion sessions. Previous studies are properly cited and acknowledged.

5.  Suggested by Reviewer 3, detailed discussion and comparison with former studies have been added to the revised manuscript as well as proposing future works that can be extended from our current study.

**Specific responses to Review 1**
The goal of this study is to extend the results of studies such as Lebsock et al. (2013) and Boutle et al. (2014) on quantifying the effects of sub-grid scale inhomogeneity on microphysical process rates applied in GCMs from observations. The central tenet is that inhomogeneity varies with length scale and meteorological regime, thus the currently standard use of "universal" constants to characterize inhomogeneity cannot adequately describe subgrid-scale variability across a range of horizontal grid sizes or environmental conditions. The authors use a temporally extensive remote sensing dataset primarily sampling shallow convection over Graciosa Island in the Azores to develop "scale-aware" enhancement factors for the autoconversion and accretion processes ($E_{auto}$ and $E_{accr}$, respectively) for several commonly used bulk microphysical parameterizations. These enhancement factors are estimated from compositing of variances and covariances of instantaneous retrievals of cloud and rain liquid water path (CLWP and RLWP, respectively) and cloud drop number concentration Nc over varying time windows, which the authors argue are roughly equivalent to a GCM horizontal grid length if a constant wind speed is assumed.

Thank you for the comments.
As stated in general response, we now use collocated $q_c$ and $q_r$ in the calculations and we use wind speed from merged sounding over certain periods to mimic model grid size.

I agree with the authors' basic premise that the use of constant values for $E_{auto}$ and $E_{accr}$ in GCM microphysics schemes is unrealistic and likely introduces precipitation biases similar (perhaps in magnitude if not sign) to assuming that grid-mean quantities (e.g. of $N_c$ and cloud and rain liquid water mixing ratios $q_c$ and $q_r$) are applicable to calculation of process rates in models with coarse grids (say horizontal grid length L greater than a kilometer or so). Furthermore, their assertion that enhancement factors should vary as a function of L as well as meteorological regime is well-stated, although they are not able to access independent information on aerosol-cloud interactions, which I suspect may be of comparable importance to the stability and LWP criteria analyzed.

Thank you for the comments.
In the revised manuscript, we keep the part of assessing enhancement factors for different grid sizes and add the uncertainties in enhancement factor calculations came from our retrieval uncertainties (Figure 4).

We agree that, within the same meteorological regime and similar LWP, aerosol-cloud-precipitation interactions will affect sub-grid cloud and precipitation variabilities. However, it is a challenge to quantitatively estimate this effect using our existing dataset, especially with large uncertainties in aerosol measurements during drizzling conditions. This is an interesting topic and worth to explore in the further.

For completeness and clarification, we add following to lines 488-490 in the revised manuscript: "The effect of aerosol-cloud-precipitation-interactions on cloud and precipitation sub-grid variabilities may be of comparable importance to meteorological regimes and precipitation status and deserves a further study."

Despite agreeing with the importance and timeliness of the premise of the manuscript, I have several major issues with the relevance of the observations to diagnosis of microphysical process inhomogeneity. Most importantly, the retrievals of cloud and rain/drizzle properties are not collocated; drizzle properties are only retrieved below cloud base. Cloud and drizzle properties are convolved within cloud such that what is classified as CLWP in fact includes contributions from in-cloud drizzle as well. Microphysical process rate equations assume coincident cloud and rain water mixing ratios (accretion) and coincident cloud water and drop number concentration (autoconversion), so unless it could be shown from some other dataset (LES? Aircraft observations? Maybe even a simplified 1D model?) that subcloud RLWP correlates highly with in-cloud RLWP and has similar magnitude, I have serious doubts about the physical relevance of the retrieved covariances. This may explain the apparently low ratios of cloud to rain water presented in the paper (see lines 33-34 and 291-293, Fig. 2e-f), although the authors give no "expected" value of this ratio for comparison.

Thanks for your comments and suggestions.
In the revised manuscript, collocated joint retrieval of cloud and drizzle LWC is employed to obtain $q_c$ and $q_r$ simultaneously. We updated the calculations accordingly, now using the variance and covariance of in-cloud mixing ratios.
In Figures 2e and 2f, we superimpose the ratio of layer-mean $q_r$ to $q_c$ and the ratios are both less than 15% in the two panels. This is also evident in Figure 1b that 10 times of $q_r$ is still less than $q_c$. The differences in magnitude are consistent with previous study (e.g., CloudSat and aircraft measurement presented by Boutle et al. 2014, their Figure 1a).
We add the following sentences to lines 332-334 in the revised manuscript: "In both panels, the ratios are less than 15%, which means that $q_r$ can be one order of magnitude smaller than $q_c$. The differences in magnitude are consistent with previous CloudSat and aircraft results (e.g., Boutle et al. 2014)."

The use of column-integrated liquid water paths introduces further uncertainty because the partitioning of the collision-coalescence process into autoconversion and accretion sub-processes is heterogeneous in the vertical. In the shallow clouds typical of the ENA site, autoconversion will be dominant near cloud top where cloud droplets have reached a maximum size due to condensation and larger drizzle drops are rare while accretion dominates lower in cloud, where the drizzle drops initially formed at cloud top sediment and continue to grow by collecting cloud droplets. Erasing this coherent vertical variability by the use of integrated water paths may bias the results presented: in stratiform clouds, liquid water is at a maximum near cloud top (i.e. CLWP is weighted toward cloud top), such that the $E_{accr}$ values in particular are using over-inflated liquid water values.

Thanks for your comments and suggestions.
We agree that autoconversion and accretion sub-processes dominate at different levels of cloud and it is physically reasonable to calculate them separately using different parts of the $q_c$ and $q_r$ profiles.

Following your suggestion, we add the followings to methodology part in lines 220-229 in the revised manuscript: "The autoconversion and accretion parameterizations partitioned from collision-coalescence process dominate at different levels in a cloud layer. Autoconversion dominates around cloud top where cloud droplets reach maximum by condensation and accretion is dominant at middle and lower parts of the cloud where drizzle drops sediment and continue to grow by collecting cloud droplets. Complying with the physical processes, we estimate autoconversion and accretion rates at different levels of a cloud layer in this study. The averaged $q_c$ within the top five range gates (~215 m thick) are used to calculate $E_{auto}$. To calculate $E_{accr}$, we use averaged $q_c$ and $q_r$ within five range gates around the maximum radar reflectivity. If the maximum radar reflectivity appears at the cloud base, then five range gates above the cloud base are used."

I'm also confused about how the authors transformed liquid water paths to mixing ratios. They state that "CLWC [cloud liquid water content] values are transformed to $q_c$...by dividing by air density" (lines 191-192) and similar for qr (lines 194-195) but never define how they calculate CLWC or drizzle LWC. Are they dividing water path by cloud/drizzle shaft depth for an average value? Or are they applying the methods of Xie and Zhang (2015) and Wu et al. (2015) to the retrievals?

Thank you for the comments.
We first retrieve CLWC and RLWC profiles, then divided by air density vertical profiles calculated from temperature and pressure in merged sounding.

For clarification, we add the following sentences in the revised manuscript: "Using air density ($\rho_{air}$) profiles calculated from temperature and pressure in merged sounding, mixing ratio ($q$) can be calculated from LWC using $q(z) = LWC(z)/\rho_{air}(z)$." to lines 213-214 and 559-561 in methodology and Appendix A.

Is the retrieval of $N_c$ vertically resolved? This part of the methodology is insufficiently described to understand what the authors did, and regardless, it doesn't address the issue that drizzle properties can only be retrieved below cloud using their approach.

Thank you for the comments.
In our study, $N_c$ is not vertically resolved but is assumed to be constant in a cloud layer.
For clarification, the following is added to lines 208-209 in methodology part in the revised manuscript: "Cloud droplet number concentration ($N_c$) is retrieved using the methods presented in Dong et al. (1998, 2014a and 2014b) and are assumed to be constant in a cloud layer".
The drizzle properties in the revised manuscript are not from below cloud only, instead, $q_r$ is now vertically resolved.

Finally, the authors made no attempt to quantify the uncertainty of the reported enhancement factors, such that I cannot make a determination as to whether their $E_{auto}$ and $E_{accr}$ are statistically distinct from the constant values introduced by Morrison and Gettelman (2008). This is particularly relevant to Figure 4.

Thanks for your comments.
To assess the uncertainty associated with the retrieved $q_c$ and $q_r$, we vary $q_c$ and $q_r$ within their corresponding uncertainties, e.g., $(1 \pm 0.18)q_d$ and $(1 \pm 0.3)q_c$ and re-do the calculations. The mean differences are used as the boundaries of $E_{auto}$ and $E_{accr}$ as shown in Figure 4 in the revised manuscript.

We add the following sentences to lines 215-219 to address the uncertainties of $E_{auto}$ and $E_{accr}$: "The estimated uncertainties for the retrieved $q_c$ and $q_r$ are 30% and 18%, respectively (see Appendix A). We used the estimated uncertainties of $q_r$ and $q_c$ as inputs of Eqs. (4) and (7) to assess the uncertainties of $E_{auto}$ and $E_{accr}$. For instance, $(1 \pm 0.3)q_c$ are used in Eq. (4) and the mean differences are then used as the uncertainty of $E_{auto}$. Same method is used to estimate the uncertainty for $E_{accr}$."

Also, in the discussion of Figure 4, we add the following sentences to lines 396-340 in the revised manuscript "The shaded areas represent the uncertainties of $E_{auto}$ and $E_{accr}$ associated with the uncertainties of the retrieved $q_c$ and $q_r$. When model grid increases, the uncertainty slightly decreases. The prescribed $E_{auto}$ is close to the upper boundary of uncertainties except for the 30-km grid, while the prescribed $E_{accr}$ is significantly lower than the lower boundary."

I would also have liked to see the authors show the quantitative impact of treating qc and $N_c$ individually with respect to calculating Eauto, as their derivation of Equation 4 assumes that the covariability of qc and $N_c$ can be ignored. While the magnitude of Eauto is comparable for $q_c$ or $N_c$ individually, I don't have a good sense for what including variability of both variables implies for the predicted $E_{auto}$ values. It's certainly a problem that CLWP and $N_c$ are correlated in the ARM dataset employed, but that doesn't change the fact that variability of $N_c$ is likely substantial, especially for the longer time periods analyzed or in more cumuliform precipitation.

Thank you for the comments.
Due to $N_c$ and LWP are highly correlated in our retrieval algorithm, we are currently unable to assess the covariance of $q_c$ and $N_c$ in autoconversion parameterization. In other words, the Nc is derived from LWP and other cloud variables ($r_e$ and cloud thickness). We can use the following two figures to show why these results are artificially high: the $E_{auto}$ calculated from the covariance of $N_c$ and $q_c$ for 60-km (left panel) and 180-km grid (right panel) sizes superimposed by average precipitation frequency in each bin can reach 40-50. Therefore, we only assess the individual effect of $N_c$ as shown in Figures 2c and 2d, which are similar to the effect of $q_c$ as shown in Figures 2a and 2b. For simplicity and clarity, only $E_{auto}$ calculated from $q_c$ are included in the discussions afterword.

[Figure]

[Figure]

For clarification, we add the following sentences to lines 321-325 in the revised manuscript "Because the $E_{accr}$ values calculated from $q_c$ and $N_c$ are close to each other, we will focus on analyzing the results from $q_c$ only for simplicity and clarity. The effect of $q_c$ and $N_c$ covariance, as stated in Section 4.1, is not presented in this study due to the intrinsic correlation in the retrieval (Dong et al., 2014a and 2014b and Appendix A of this study)."

In light of these concerns, I must recommend that this manuscript be **rejected** in its current form. A revised version of the manuscript only addressing autoconversion would be more feasible and would also be very useful to the parameterization development community, although as mentioned above, I would ask that the authors address the question of whether ignoring covariability of qc and $N_c$ is a reasonable assumption.

Thank you for the comments.
In fact, we found very high covariance between the two variables, which is a result of our retrieval method in which $N_c$ is derived from LWP and other cloud variables. As stated in the response to last comment, the results using $N_c$ and $q_c$ covariance could result in large variations of $E_{auto}$ that are artificially high. To address this issue, independent retrieval methods for $N_c$ and $q_c$ are needed, that is what we plan to explore in the future.

Thanks for suggesting to use the jointly retrieved $q_c$ and $q_r$, we think it is reasonable to keep accretion part in the manuscript.

I would be happy to review a revised and refocused manuscript. Until remote sensing datasets can unambiguously partition in-cloud condensed water into cloud and drizzle components, analysis of cloud-rain covariance from the present spatially disjoint cloud and rain retrievals cannot be used to inform accretion parameterizations.

Thank you for the comments.
Please see above responses that we tried to retrieve $q_c$ and $q_r$ profiles in the cloud and re-do the calculations.

A technique like that of Luke and Kollias (2013; doi:10.1175/JTECHD-11-00195.1) that uses skewness of the Doppler spectrum to differentiate between cloud and drizzle could be combined with a method similar to Frisch et al. (1998; doi:10.1029/98JD01827) to retrieve vertically-resolved profiles of cloud and rain water, albeit likely only in stratiform clouds. If such an approach could be developed, the analysis performed in this manuscript would be more tractable although it would likely need to be validated before application to the GCM cloud inhomogeneity problem given the amount of technical work necessary to provide confidence in the retrievals.

Thank you for the suggestions.
We used an alternative way as presented in Appendix A to retrieve CLWC and RLWC and then calculate $q_c$ and $q_r$. The uncertainties of the retrieval are difficult to quantify without aircraft *in situ* data or other retrieval results. In the uncertainty analysis part, we used 18% as uncertainty for RLWC (rain LWC) from drizzle properties in Wu et al. (2015) and 30% for CLWC (cloud

LWC) from cloud properties in Dong et al. (2014a and 2014b). The actual uncertainties may vary depend on the accuracy of merged sounding data and WACR detectability near cloud base.

In Appendix A, we add the following sentences to address the retrieval uncertainties to lines 545-554: "It is difficult to quantitatively estimate the retrieval uncertainties without aircraft in situ measurements. For the proposed retrieval method, 18% should be used as uncertainty for RLWC from drizzle properties in Wu et al. (2015) and 30% for CLWC from cloud properties in Dong et al. (2014a and 2014b). The actual uncertainty depends on the accuracy of merged sounding data, the detectability of WACR near cloud base and the effect of entrainment on cloud adiabaticity during drizzling. In the recent aircraft field campaign, the Aerosol and Cloud Experiments in Eastern North Atlantic (ACE-ENA) was conducted during 2017-2018 with a total of 39 flights over the Azores, near the ARM ENA site on Graciosa Island. These aircraft in situ measurements will be used to validate the ground-based retrievals and quantitatively estimate their uncertainties in the future."

References:
Boutle, I. A., Abel, S. J., Hill, P. G., and Morcrette, C. J.: Spatial variability of liquid cloud and rain: Observations and microphysical effects. Quart. J. Roy. Meteor. Soc., 140, 583–594, doi:10.1002/qj.2140, 2014.

Dong X., Ackerman, T. P., and Clothiaux, E. E.: Parameterizations of Microphysical and Radiative Properties of Boundary Layer Stratus from Ground-based measurements, J. Geophys. Res., 102, 31,681-31,393, 1998.

Dong, X., Xi, B., Kennedy, A., Minnis, P. and Wood, R.: A 19-month Marine Aerosol-Cloud_Radiation Properties derived from DOE ARM AMF deployment at the Azores: Part I: Cloud Fraction and Single-layered MBL cloud Properties, J. Clim., 27, doi:10.1175/JCLI-D-13-00553.1, 2014a.

Dong, X., Xi, B., and Wu, P.: Investigation of Diurnal Variation of MBL Cloud Microphysical Properties at the Azores, J. Clim., 27, 8827-8835, 2014b.

Wu, P., Dong, X. and Xi, B.: Marine boundary layer drizzle properties and their impact on cloud property retrieval, Atmos. Meas. Tech., 8, 3555–3562. doi: 10.5194/amt-8-3555-2015, 2015.

**Specific responses to Review 2**

This paper discusses how variability of cloud and rain at the GCM sub-grid scale affect the parametrizations of autoconversion and accretion that are typically used. This has become a popular topic in recent years with many papers and modelling centres using this as a method of improving warm rain simulation. The current paper has some novel aspects, for example the use of data from the Azores to evaluate parametrizations, but I feel would require some significant modifications before it is acceptable for publication.

We made the point-to-point response and thank for your suggestions that help us a lot improve the manuscript. We appreciate the references that you provided and add them in the revision.

Major comments:

1. I don't feel this paper fully or correctly acknowledges the previous work that has been done in this field, which leads to many statements with are either misleading, incorrect, in contradiction to previous studies without explanation, or presented as new when actually they have been published before. Specific examples of this are:

Thanks for your comments and suggestions on literature review. We have revised the sentences and properly acknowledged previous studies.

a) L31, 284, 390 and elsewhere - repeatedly the authors refer to "GCMs", implying that they are stating a common feature of many models, whereas in actual fact they are referring specifically to the MG08 microphysics scheme which is only used in a very small number of GCMs. This terminology needs to be more precise, to highlight the fact that not all GCMs make the same assumptions as MG08.

Thank you for the comment.

The terminology has changed in the revision, we mainly used MG08 scheme in the calculation and discussion. We also give the values for 60-km and 150-km grid sizes for other parameterizations listed in Table 1. Same approaches can be repeated for other parameterization schemes used in GCMs.

To avoid confusion, we add the following sentences at the end of the introduction: "Most of the calculations and analyses in this study is based on Morrison and Gettleman (2008, MG08 hereafter) scheme. The enhancement factors in several other schemes are also discussed and compared with the observational results and the approach in this study can be repeated for other microphysics schemes in GCMs." in lines 126-130 in the revised manuscript.

b) L99 - this statement is incorrect - whilst some models do use prescribed values regardless of meteorological conditions, the whole point of Boutle et al (2014), which is cited as introduction to this statement, is to provide a parametrization depending on meteorological conditions which can be used in GCMs. This parametrization is improved upon by Hill et al (2015), who add in a regime dependence to the parametrization, and implemented in a model by Walters et al (2017). The authors need to acknowledge this work in the context of their own.

Thanks for the correction.

This part has been rephrased to "Boutle et al. (2014) used aircraft in situ measurements and remote sensing techniques to develop a parameterization for cloud and rain, in which not only consider the sub-grid variabilities under different grid scales, but also consider the variation of cloud and rain fractions. The parameterization was found to reduce precipitation estimation bias significantly. Hill et al. (2015) modified this parameterization and developed a regime and cloud type dependent sub-grid parameterization, which was implemented to the Met Office Unified Model by Walters et al. (2017) and found that the radiation bias is reduced using the modified parameterization." In lines 99-106 in the revised manuscript.

c) L293-294 - this statement is just repeating the previous conclusions of Boutle et al (2014) and Lebsock et al (2013).
Thank you for the comment.
The two studies are cited and acknowledged in the context of our results.

d) L335 - Hill et al (2015) also show regime dependence and should be cited here.
Thanks for the comment.
The sentence is rephrased to "Therefore, as suggested by Hill et al. (2015), the selection of $E_{auto}$ and $E_{accr}$ values in GCMs should be regime-dependent." in lines 382-383 in the revised manuscript.

e) L336-337 - I don't understand this statement - why is it difficult to vary enhancement factors in GCMs? Walters et al (2017) using the parametrizations of Boutle et al (2014) does exactly that - there is nothing difficult here and no reason why other GCMs could not do similar.

Thanks for the comment.
We deleted this sentence and rephrased this part to "To properly parameterize sub-grid variabilities, the approaches by Hill et al. (2015) and Walters et al. (2017) can be adopted. To use MG08 and other parameterizations in GCMs as listed in Table 1, proper adjustments can be made according to the model grid size, boundary layer conditions, and precipitating status." in lines 384-387 in the revised manuscript.

f) L364-368 - I don't fully understand what is being claimed here, and it certainly is not supported by any evidence presented in the paper. But what I think the authors are saying is that in more cumulus-type (less stratiform) clouds, E_auto should be smaller. This appears contradictory to the results of Boutle et al (2014) (their Fig 10) and Hill et al (2015) which show that E_auto is higher in convective type cloud regimes. It also appears in contradiction to the authors own statement on L429-430 (a statement that appears with no justification or background), that unstable boundary layers give rise to larger E_auto values. Please clarify this.

Thanks for the comment.
For this statement, we are trying to say for the 'cloud type under this study' e.g., MBL stratocumulus in this study, the $E_{auto}$ values should be smaller over land than that over ocean. To avoid confusion, we deleted this statement.

The statement in L429-430 of original manuscript "The $E_{auto}$ values in both stable and mid-stable boundary layer conditions are smaller than the prescribed value of 3.2 used in GCMs, while those values in unstable boundary layers conditions are significantly larger than 3.2 regardless of whether or not the cloud is precipitating." is the conclusion we draw from the values in Table 2

where the boundary layer is classified into three categories using lower tropospheric stability (LTS). For clarification, we added '(Table 2)' at the end of this statement. This statement is now in lines 479-482 in the revised manuscript.

g) L433 - as is done in Hill et al (2015) and Walters et al (2017).

Thanks for this comment.
The sentence is rephrased to "Therefore, the selection of $E_{auto}$ and $E_{accr}$ values in GCMs should be regime-dependent, which also has been suggested by Hill et al. (2015) and Walters et al. (2017)" in lines 483-484 the revised manuscript.

h) Fig 4 - despite the constant criticism of MG08 for using a fixed value of E_auto=3.2, this figure shows that at larger grid sizes, this value is actually incredibly good – some credit should be given to MG08 for this!

Thank you for the comment.
Yes, $E_{auto}$ prescribed in MG08 is getting more and more close to those calculated from observations. In the revised Figure 4, the mean value from observation is exactly the same for 180 km model grid.

We rephrased the following "After that, the $E_{auto}$ values remain relatively constant of ~3.18 when the model grid is 180 km, which is close to the prescribed value of 3.2 used in MG08. This result indicates that the prescribed value in MG08 represents well in large grid sizes in GCMs" in lines 391-394 in the revised manuscript.

2) L148, L151 - equations 4 and 5 are incorrect, the term in the denominator should be Gamma(nu) not Gamma(a) as written (see Eq 7 of Boutle et al (2014) or Eq 6 of Pincus and Klein (2000)). I hope this is only a typo and not a problem with all of the data analysis! Also, I'm confused about whether or not you are investigating variability of Nc - the text seems to suggest you are, but this equation ignores any variability in Nc - please clarify the text and correct the equation if necessary.

Thanks for the corrections. These are typos and have been corrected. All the calculations and analysis in the original and revised manuscripts used the correct formulas.

We indeed include $N_c$ in the calculation and the equation has been changed in the revision. We only assess the individual effect of $N_c$ to $E_{auto}$, not for the covariance of $q_c$ and $N_c$ because $q_c$ and $N_c$ are highly correlated in the retrieval method and it is difficult to tell if the results are due to natural variability or due to mathematics in the retrieval. We can use the following two figures to show why these results are artificially high: the $E_{auto}$ calculated from the covariance of $N_c$ and $q_c$ for 60 km (left panel) and 180 km grid (right panel) size superimposed by average precipitation frequency in each bin can reach 40-50. Therefore, we only assess the individual effect of $N_c$ as shown in Figures 2c and 2d, which are similar to the effect of $q_c$ as shown in Figures 2a and 2b. For simplicity and clarity, only $E_{auto}$ calculated from $q_c$ are included in the discussions afterword.

[Figure]

[Figure]

For clarification, we added the following to lines 321-325 in the revised manuscript "Because the $E_{accr}$ values calculated from $q_c$ and $N_c$ are close to each other, we will focus on analyzing the results from $q_c$ only for simplicity and clarity. The effect of $q_c$ and $N_c$ covariance, as stated in Section 4.1, is not presented in this study due to the intrinsic correlation in the retrieval (Dong et al., 2014a and 2014b and Appendix A of this study).".

3) L207, 340 - simply using a constant wind speed is quite crude - most previous studies with ground based equipment (eg. Boutle et al 2014) have either used actual wind speeds or model derived reanalysis wind speeds to construct spatial scales from time averages. At the very least this simplification needs to be noted and possible errors due to this discussed.

Thank you for the comment and suggestion.
We agree that use actual wind speeds or model derived wind speeds can reduce the sampling uncertainty. In the revised manuscript, we use the averaged wind speed within cloud layer to mimic different model grid sizes.

The following paragraph is added to lines 238-245 in the methodology part: "To evaluate the dependence of autoconversion and accretion rates on sub-grid variabilities for different model spatial resolutions, an averaged wind speed within cloud layer was extracted from merged sounding and used in sampling observations over certain periods to mimic different grid sizes. For example, two hours of observations corresponds to a 72-km grid box if mean in-cloud wind speed is 10 $m\ s^{-1}$ horizontal wind and if the wind speed is 5 $m\ s^{-1}$, four hours of observations is needed to mimic the same grid. We used six grid sizes (30-, 60-, 90-, 120-, 150-, and 180-km) and mainly show the results from 60-km and 180-km grids in Section 4."

4) L220 onwards, L281, elsewhere - the analysis appears to be presented in terms of LWP and RWP, i.e. column integrals of quantities. This is very different to the LWC and RWC, i.e. grid-box mean quantities which are used in parametrizations. Most previous studies have used LWC and RWC to calculate the variability, and so the results are directly applicable to parametrizations. It's not clear to me that results presented in LWP and RWP are so directly applicable. The authors need to investigate how applicable their results using column-integral quantities are to previous studies and parametrizations - it appears from the text that you do have direct observations of LWC and RWC, so it should not be too difficult to make this comparison, or re-do the analysis using the LWC and RWC data.

Thanks for the comment.
We agree that the use of CLWP and RLWP ignores the heterogeneity of collision-coalescence process in the cloud layer. In the revised manuscript, $q_c$ and $q_r$ are jointly retrieved and applied to the calculation.

We add the following sentences to methodology part lines 220-229 in the revised manuscript: "The autoconversion and accretion parameterizations partitioned from collision-coalescence process dominate at different levels in a cloud layer. Autoconversion dominates around cloud top where cloud droplets reach maximum by condensation and accretion is dominant at middle and lower parts of the cloud where drizzle drops sediment and continue to grow by collecting cloud droplets. Complying with the physical processes, we estimate autoconversion and accretion rates at different levels of a cloud layer in this study. The averaged $q_c$ within the top five range gates (~215 m thick) are used to calculate $E_{auto}$. To calculate $E_{accr}$, we use averaged $q_c$ and $q_r$ within five range gates around the maximum radar reflectivity. If the maximum radar reflectivity appears at the cloud base, then five range gates above the cloud base are used."

General comments:
Title - should probably be "Evaluation of ..."
Thanks. Title has been changed.

L50 - should say "a significant amount of drizzle is evaporated"
Thanks. This has been changed.

L56 - I'm not entirely sure I agree with this statement - change in albedo (i.e. the first indirect effect) is the most significant indirect effect. There is also an extensive literature on buffering of the 2nd indirect effect and mechanisms through which aerosol could even enhance convective precipitation. At the very least this statement needs to be more accurate in the context it is being used - increases in aerosol are mainly thought to suppress precipitation in MBL clouds.
Thanks for the comment and sorry for the confusion.
This sentence means that the aerosol indirect effect associated with MBL cloud constitutes the major part in global aerosol indirect effect.
To avoid confusion, this sentence is deleted in the revised manuscript.

L62 - MG08 is an odd reference here, given it discusses a microphysics parametrization, something which is required in models of all scales
Thank you for the comment. This reference is deleted in the revised manuscript.

L63 - the "process" of autoconversion and accretion only exist because modellers have partitioned the liquid water into "cloud" and "rain" categories - please rephrase this sentence, they are not real processes, all that happens in the real atmosphere is collisioncoalescence of water droplets.
Thanks for the comment.
This sentence is rephrased to "For Example, warm rain parameterizations in most GCMs treat the condensed water as either cloud or rain in the process of collision-coalescence, which is partitioned into autoconversion and accretion sub-processes in model parameterizations"

L64, 72, 73, 122, 129 - the references to MG08 and LG13 are odd here, given they do not propose autoconversion or accretion parametrizations of their own, they use the scheme of KK00 which is already referenced.
Thank you for the comment. The odd references have been deleted.

L77 - using a prime to denote grid-mean quantities is somewhat non-standard – an overbar is the more typical symbol for a mean quantity.
Thank you for the suggestion. The symbols have been changed in the equations and text.

L79 - I'm not sure I follow why positive skewness is important - can you elaborate? It is only really the non-linear form of the equations that mean rates depend strongly on the sub-grid variability.

Thanks for the comment.
The skewness determines the degree of error by using mean value to represent entire domain. If LWP is normally distributed then mean value is equal to mode value, meaning that mean value can represent the value that most frequently occurs in the field. Whereas in skewed distributions, e.g., Gamma distribution where mean value is greater than mode, then the mean value only represents a relatively small portion of the samples. And using mean to represent entire field results in larger errors than that in normal distribution.

For clarification, we rephrased this sentence to "MBL cloud liquid water path (CLWP) distributions are often positive skewed (Wood and Hartmann, 2006; Dong et al. 2014a and 2014b), that is, the mean value is greater than mode value. Thus, the mean value only represents a relatively small portion of samples. Also, due to the nonlinear nature of the relationships, the two processes depend significantly on the sub-grid variability and co-variability of cloud and precipitation microphysical properties" in lines 77-81 in the revised manuscript

L100 - Boutle et al (2014) use a combination of aircraft, ground-based and satellite measurements.
Thanks for the comment. The citation to this reference has been changed to "Boutle et al. (2014) used aircraft in situ measurements and remote sensing techniques to develop a parameterization for cloud and rain, in which not only consider the sub-grid variabilities under different grid scales, but also consider the variation of cloud and rain fractions. The parameterization was found to reduce precipitation estimation bias significantly." In lines 99-103 in the revised manuscript.

L312 - using flash flooding as an example when discussing drizzling marine stratocumulus is a bit of a leap, I suggest removing this statement unless you have any evidence that extreme rainfall rates are affected.
Thank you for the comment.
The flash flooding is not a suitable example in the context of marine stratocumulus. We have rephrased this sentence to: "providing limited information in estimating rain water evaporation and air-sea energy exchange" in lines 360-361 in the revised manuscript.

This manuscript uses ground-based observations and retrievals from the DOE ARM Mobile Facility at Graciosa Island over the Azores to calculate and characterize shallow cloud autoconversion and accretion enhancement factors as a function of temporal and spatial size, which help establish the observational benchmarks with which to compare against autoconversion and accretion factors in GCM parameterizations. This is a worthy goal, as many GCMs precipitate both too frequently and too lightly relative to observations, with the preliminary inference that the sub-grid enhancement factor for autoconversion in many parameterizations is too strong, and the factor for accretion is too weak. This is because autoconversion primarily determines precipitation initiation, and accretion primarily controls drizzle/precipitation intensity. The approach taken in this study both interesting and instructive, as the homogeneity of cloud/liquid water path (LWP) properties tends to be higher for smaller grid sizes, suggesting that GCMs with finer resolutions many have an even weaker accretion enhancement factor than coarser models. This appears to be an important consideration for models with a range of grid resolutions. While autoconversion and accretion biases are systematically characterized in this study, the additional novelty of this work is that the biases are not fixed, but rather regime dependent, with both lower tropospheric stability and precipitation playing a significant role.

Thanks for the comments.

The layout and results of this study have the potential to inform existing and future parameterizations about how to tailor precipitation enhancement factors as a function of local thermodynamics, resolution, temporal length, and precipitation itself. While other regions will need to be studied as well to gather and calculate additional autoconversion and accretion enhancement factors in other regimes and/or large-scale dynamics, this study is a good start in providing potential guidance for the modeling and model analysis communities. While there is much potential in this manuscript, there are also a number of minor to moderate technical and science questions which need addressing prior to consideration for publication, and these are mentioned below.

Thanks for the comments. Please see the point-to-point response below.

Though the manuscript was not intended to be an exhaustive study of all the factors that that may modulate the autoconversion and accretion enhancement factors, it might be complementary to the study to also consider a few additional large-scale factors, such as local vertical velocity profiles, as has recently been done in work examining entrainment velocities over the MAGIC campaign over the Northeast Pacific, in which even during the boreal summer, approximately 20% of the profiles were observed to have rising motion near cloud top. Whether rising motion would enhance accretion rates/enhancement factors might complement the findings presented in this study.

Thanks for the comments and suggestions.
Vertical velocity within cloud layer can be inferred from cloud radar Doppler velocity. However, the drizzle presents ~46% of the time at Azores site, thus the vertical velocity signal is contaminated by the presence of drizzle. Unfortunately, we do not have a reliable retrieval method to determine the reliable vertical velocity from Doppler velocity. Based on your suggestion, we have proposed this work in the summary and future work session in the revised manuscript in lines 488-494: 'The effect of aerosol-cloud-precipitation-interactions on cloud and precipitation sub-grid variabilities may be of comparable importance to meteorological regimes and precipitation status and deserves a further study. Other than the large-scale dynamics, e.g., LTS in this study, upward/downward motion in sub-grid scale may also modify cloud and precipitation development and affect the calculations of enhancement factors. The investigation of the dependence of $E_{auto}$ and $E_{accr}$ on aerosol type and concentration as well as on vertical velocity would be a natural extension and complement of current study.'

Finally, at the end of this review is an enumeration of grammatical suggestions/typos; the list is non-exhaustive such that a thorough proofreading will be essential prior to publication.

Thanks for catching the errors/typos. They have been corrected and a thorough proofreading is performed.

**Science/Technical Questions and Comments:**
1) On lines 33-34 (and also lines 291- 293), the authors state that "the ratios of rain to cloud liquid water at Eaccr=1.07 and Eaccr=2.0 are 0.048 and 0.119, respectively, further proving that the prescribed value of Eaccr=1.07 used in GCMs is too small to simulate precipitation intensity", but it is somewhat unclear to me how this proves this, unless the authors include (for clarity) the respective GCM RLWP-to-CLWP ratios as well in this statement.

Thanks for the comment.
The ratio of rain to cloud water mixing ratio at $E_{accr}$=1.07 is 0.063 in the revised manuscript, and this ratio keeps increasing until ~0.142 at around $E_{accr}$=2. The ratio between rain to cloud water mixing ratio at $E_{accr}$=1.07 is substantially smaller than that at $E_{accr}$=2, indicating that the fraction of rain water in total water is too low at $E_{accr}$=1.07. A higher $E_{accr}$ value is accompanied by a higher ratio, which indicates an increased precipitation intensity. Since we did not perform a sensitivity study using GCM simulation in this study, it is hard for us to include GCM RLWP-to-CLWP ratios.
For clarification, this sentence is rephrased to 'The ratio of $q_r$ to $q_c$ increases from $E_{accr}$=1.07 (0.063) to $E_{accr}$=2.0 (0.142), indicating that the fraction of rain water in total water using the prescribed $E_{accr}$ is too low. This ratio could be increased significantly using a large $E_{accr}$ value, therefore increasing precipitation intensity in the models. This further proves that the prescribed value of $E_{accr}$=1.07 used in MG08 is too small to correctly simulate precipitation intensity in the models.' in lines 340-343 in the revised manuscript.

How does this range of ratios compare to those from Lebsock et al. (2011)? Based on their Figure 6, it seems that the RLWP-to-CLWP ratio is generally higher than 0.119, though of course other factors are work as well (e.g. cloud top effective radius, included in their Figure 8). If I am interpreting these differences correctly, what do the authors attribute to the apparently higher ratios in that observational study? Reference: Lebsock, M. D., T. S. L'Ecuyer, and G. L. Stephens, 2011: Detecting the ratio of rain and cloud water in low-latitude shallow marine clouds. J. App. Meteor. Climatology, 50, 419-432, doi: 10.1175/2010JAMC2494.1.

Thanks for the comment.

The ratios in the original manuscript are lower than those presented by Lebsock et al. (2011), because we only included RLWP below cloud base. In the revised manuscript, vertical profiles of $q_v$ and $q_r$ are retrieved and the ratios shown in Figures 2e-2f now are the ratios of layer-mean $q_r$ to $q_v$, which are generally smaller than 0.1 (more than one order of magnitude lower). This is consistent with the CloudSat and aircraft measurements presented by Boutle et al. 2014 (their Figure 1a). The averaged RLWP-to-CLWP ratio from the partitioned rain and cloud retrieval is 0.21 (not included in the manuscript), which agrees reasonably well with those in Lebsock et al. (2011).

2) Lines 77-78 (and elsewhere): The authors use primes (') to denote grid means, and are consistent about this, but wouldn't an overbar typically denote grid mean values? Often, but not always, primes are designated for deviations from the mean. Overall, this is a fairly minor comment.

Thanks for the comment. The grid mean quantities are represented by overbars in the revised manuscript. Corresponding text and equations have been changed.

3) Lines 144-145, and more generally the implications for the findings of this study: Is the shape parameter, somewhat analogous to the LWP homogeneity from Wood and Hartmann 2006 (which was simply the squared quantify of the ratio of the mean LWP to its standard deviation)? In that study, precipitation was not explicitly examined, but instead the brokenness of cloud fields was found to increase (total CF decrease) with decreasing LWP homogeneity. The greater shallow convective cellular structure with decreasing LWP homogeneity, however, may have been more conducive for heavier precipitation. In this study, the relationship appears to be the link between reduced LWP homogeneity or other cloud field homogeneity and an increase in precipitation intensity. The latter is often too small in climate models, and too homogeneous of fields may be the culprit. Perhaps even though the authors have cited Barker et al. (1996); Pincus et al. (1999), and Wood and Hartmann (2006), an even stronger parallel/analogy should be made between the objectives and findings here and those from those studies, particularly in the discussion section later on as appropriate.

Thanks for the comment.

The shape parameter in this study is calculated in the same way as in Wood and Hartmann (2006).

The following paragraph is added in the discussion in lines 279-285 in the revised manuscript: 'Using the LWP retrieved from the Moderate Resolution Imaging Spectroradiometer (MODIS) as an indicator of cloud inhomogeneous, Wood and Hartmann (2006) found that when clouds become more inhomogeneous, cloud fraction decreases, and open cells become dominant with stronger drizzling process (Comstock et al., 2007). The relationship between reduced homogeneity and stronger precipitation intensity is found in this study, which is similar to the findings in other studies (e.g., Wood and Hartmann, 2006, Comstock et., 2007, Barker et al., 1996; Pincus et al., 1999).'

3) Results in Figure 1: Traditionally, we think of the large number concentrations decreasing with precipitation, as heavier precipitation is dominated by fewer, larger drops. It perhaps seems a little surprising that a peak (red bin denoting Nc) emerges during the heavy drizzle phase of Nc values of 100 cm^-3 (Figure 1d). However, there are fewer very high values of Nc (e.g. >150 cm^-3 ) during the hours of moderate to heavy drizzle. From Fig. 1b as well, there does seem to be high values of Nc just before or after periods of heavier drizzle, but a suppression of Nc is observed somewhat during the stronger pulses of precipitation. Can the authors discuss perhaps why more of the larger number concentrations are not removed during the heavier drizzle events? On the other side of the spectrum, during most of the non-precipitating phase, there is a local maximum of Nc at 150 cm^-3. Have the authors considered looking at the corresponding time series of effective radius to include as perhaps another panel for Figure 1? This may complement the Nc observations quite nicely.

Thanks for the comments.
During the drizzling period (1d), the microwave radiometer retrieved cloud LWP, as well as retrieved $r_e$ and $N_c$ are dominated by cloud droplets, not drizzle. Therefore, the differences in LWP and $N_c$ (as well as $r_e$) between non-drizzle and drizzling periods are not obvious in this study. From the retrieval method, $r_e$ highly depend on LWP values. Whenever LWPs are close, the retrieved $r_e$ are close. Dong et al. (2015) reported statistical results of $r_e$ and $N_c$ using six selected cases at the Azores and found that $r_e$ is 8.2-9.5 $\mu m$ in non-drizzling cloud and 11.0-18.9 $\mu m$ in drizzling cloud, $N_c$ is 119.1-145.9 cm$^{-3}$ in non-drizzling cloud and 25.1-100.3 cm$^{-3}$ in drizzling cloud. Thus, statistically, cloud particle size is larger and number concentration is lower in drizzling cloud compared with the non-drizzling counterpart. It is hard to draw conclusions from the values in one case.

Because this manuscript is not to focus on analyzing microphysical properties during precipitation and $r_e$ was not used in the calculation and analysis, we do not include $r_e$ in the figure and text.

[Figure]

4) Lines 196-197: Is this the traditional LTS definition of potential temperature at 700 hPa minus potential temperature at near the surface (or 1000 hPa)?

Yes, it is the traditional definition. To clarify, we added '$LTS = \theta_{700\ hPa} - \theta_{1000\ hPa}$' to line 231 in Section 3.

5) Figure 2: Generally, the precipitation frequency ranges from about 0.1 to just above 0.4, which seems a little low (at least the highest end of precipitation incidence) compared to other observational studies (e.g. Kubar et al. 2009). What is the definition of precipitation frequency here – is it any precipitation observed in the column, or only that which reaches the surface? Reference: Kubar, T. L., D. L. Hartmann, and R. Wood, 2009: Understanding the importance of microphysics and macrophysics in marine low clouds, Part I: satellite observations. J. Atmos. Sci., 66, 2953-2972, doi: 10.1175/2009JAS3071.1

Thanks for the comment.
The definition of precipitation frequency in Figure 2 is the average precipitation frequency of the samples in each bin and precipitation status is labelled using the criteria of maximum reflectivity below cloud base (greater than -37 dBZ as drizzling). The average precipitation frequency from Wu et al. (2015) is 47% using the same definition. Using a different definition (~ -50 dBZ at cloud base), Rémillard et al. (2012) got precipitation frequency of 70% at the same site.

Using the averaged LWP (109-140 g m$^{-2}$) and $r_e$ (12.5-12.9 μm) from Dong et al. (2014), the precipitation frequency from Figure 11 of Kubar et al. (2009) are 0.1-0.7 and 0.2-0.75 depending on the regions. The values in our study do not dependent on LWP or $r_e$ values but agree within reasonable range of those in Kubar et al. (2009).

The following sentence is added to lines 302-304 in the revised manuscript for clarification: 'Given the average LWP at Azores from Dong et al. (2014b, 109-140 g m$^{-2}$), the precipitation frequency (black lines in Figures 2a and 2b) agrees well with those from Kubar et al. (2009, 0.1-0.7 from their Figure 11).'

6) Is the E_auto critical threshold of 4 of converting cloud to drizzle drops found by the authors from the results in Figure 2 considered a novel finding, or has this been reported elsewhere as well (or perhaps a slightly different threshold)?

Thanks for the comment.
In the revised manuscript in which the $q_c$ and $q_r$ are used in calculating $E_{auto}$ and $E_{accr}$, precipitation rates have larger fluctuations than in the original manuscript. we are not able to conclude a critical value in $E_{auto}$ to clearly separate precipitation rates especially for the 180-km grid (Figure 2b). The statement of critical $E_{auto}$ was deleted from the manuscript. Lebsock et al. (2012) reported a unimodal distribution of $E_{auto}$ with a range of 1-8. Boutle et al. (2014) binned $E_{auto}$ according to cloud fraction and found that $E_{auto}$ can be as high as 4. Other than those, no studies contrasted $E_{auto}$ with precipitation frequency to our knowledge.

7) Lines 284-285 and Figure 4: The authors cite and use 1.07 as a representative value for E_accr (based on Morrison and Gettleman 2008); is this a fairly common value used in other GCM parameterizations as well? Presumably, based on this study, the range for E_accr in climate models is smaller than the observed/calculated 1-4 range for E_accr found in this study.

Thanks for the comment.
The $E_{accr}$ = 1.07 in MG08 scheme has been widely used in most GCMs. However, other microphysical parameterizations are also used in GCMs like what we listed in Table 1. For clarification, we changed the terminology in the revised manuscript that specifically evaluate the performance of enhancement factors in MG08 scheme while giving suggested values for other parameterizations (Table 3).
Also, this study is not intended to comment on a specific scheme but to provide a method that can be applied to almost every microphysical scheme using ground-based observations and retrievals.

8) Lines 354-354 and implications of study: As alluded to in the Introductory remarks of this review above, an interesting finding of this study is that the enhancement factors are even more different/biased in finer-resolution GCMs versus observations than coarser-resolution models. Thus, even though a frequent goal is improving resolution in simulations, more care is needed to address the "too frequent/too light" precipitation problem. This study appears to be instructive in how to potentially overcome this barrier.

Thanks for the comments. We highlighted the possible explanation to the 'too frequent yet too light' problem in GCM precipitation estimations in the Abstract and summary part of the manuscript.

9) Also, as alluded to in the introduction, while the autoconversion/ accretion enhancement factor dependence on both scale and LTS regime is very intriguing, the authors may also want to expand (or propose for future work) the dependence on either near cloud-top vertical velocity (e.g. from reanalysis data such as ECMWF) and/or boundary layer vertical velocity. My assumption might be that the behavior may be similar to stability; for upward motion near cloud top, both autoconversion/ accretion factors may be higher (as they are for reduced LTS), but it would be interesting to know how comparable such an effect may be to LTS.

In a somewhat similar vein, I do commend the authors for discussing that other variables (e.g. aerosol type and concentration) may be important as well for the two enhancement factors studied in this investigation in the very last paragraph. This at least sets up where the authors or others can proceed to continue to expand this line of research.

Thanks for the comments and suggestions.

The following is added to the last paragraph in lines 488-494 in the revised manuscript: 'The effect of aerosol-cloud-precipitation-interactions on cloud and precipitation sub-grid variabilities may be of comparable importance to meteorological regimes and precipitation status and deserves a further study. Other than the large-scale dynamics, e.g., LTS in this study, upward/downward motion in sub-grid scale may also modify cloud and precipitation development and affect the calculations of enhancement factors. The investigation of the dependence of $E_{auto}$ and $E_{accr}$ on aerosol type and concentration as well as on vertical velocity would be a natural extension and complement of current study.'

10) Lines 398-399:

The authors list a number of studies from the mid-2000s which discuss existing parameterizations, but are the latest GCMs quite similar as well? Is there a good recent paper or series of papers which discuss recent parameterization updates, if they exist?

Thanks for the comment.

The autoconversion and accretion parameterizations listed in this study are typical and classical schemes that are used in cloud and precipitation microphysics parameterizations. Michibata and Takemura (2015) evaluated some of the widely used autoconversion parameterizations used in GCMs and four out of five schemes in Michibata and Takemura (2015) are listed in Table 1 in our study. To our best knowledge, a recently developed parameterization is the Lee and Baik (2017) scheme, in which a physically based autoconversion parameterization was derived by solving stochastic collection equation. The collection kernel is approximated using the terminal velocity of cloud droplets and the collision efficiency is obtained from a particle trajectory model. However, the equation is much more complicated than any of the parameterizations in Table 1, we do not include Lee and Baik (2017) scheme in our study.

For completeness, we added the two references to lines 441-449 to properly acknowledge the recent studies: 'For a detailed overview and discussion of various existing parameterizations, please refer to Liu and Daum (2004), Liu et al. (2006a), Liu et al. (2004b), Wood (2005b) and Michibata and Takemura (2015). A physical based autoconversion parameterization was developed by Lee and Baik (2017) in which the scheme was derived by solving stochastic collection equation with an approximated collection kernel that is constructed using the terminal velocity of cloud droplets and the collision efficiency obtained from a particle trajectory model.

Due to the greatly increased complexity of their equation, we do not attempt to calculate $E_{auto}$ here but should be examined in future studies due to the physics feasibility of the Lee and Baik (2017) scheme.'

**Minor Notes and Grammatical Suggestions/Typos**: (*Note – this is a thorough, albeit still incomplete list of typos. Please professionally edit this manuscript prior to resubmitting.)
1) Line 28: change "increase" to "increases"
Changed.
2) Line 50: change "drizzle are" to "drizzle is"
Changed.
3) Line 62: change "Example" to "example"
Changed.
4) Line 65: change "process that drizzle drops" to "process of drizzle drops"
This sentence is rephrased.
5) Line 212: change "19-month" to "19 months"
Changed.
6) Line 221: add an "a" before "more homogeneous"
Added.
7) Line 228: add an "a" prior to "similar"
Added.
8) Line 231: change "contribute" to "contributes" for proper subject ("combination")-verb ("contributes") agreement
Changed.
9) Line 240: add "the" before "cloud"
Added.
10) Line 246: change "5-hour" to "5-hours"; similarly, for line 251: change "2-hour" to "2-hours"
The terminology is changed to '180-km grid' and '60-km grid' in the revised manuscript.
11) Line 259: add "the" prior to "autoconversion"
Added.
12) Line 273: add "a" before "similar"
This sentence is rephrased in the revised manuscript.
13) Line 317: Consider changing "seem easier to produce drizzle" to "more easily produce drizzle"
Changed.
14) Line 372: change "are representing" to "represents". Also, in general the sentence from Line 371 – 373 is slightly awkward and probably should be rewritten.
This sentence is written to 'The $q_c$ differences between models and observations are then calculated, which represent the $q_c$ adjustment in models to get a realistic autoconversion rate in the simulations.'
15) Line 378: change "associate" to "associated"
Changed.
16) Line 385: add "a" before "variety"
Added.
17) Line 659: add "are for" before "2-hr"
Changed.

[revised manuscript text omitted]
}q_c', q_r') = \dfrac{1}{2\pi \overline{q_c}q_c'\overline{q_r}q_r'\sigma_{qc}\sigma_{qr}\sqrt{1-\rho^2}} exp\left\{-\dfrac{1}{2}\dfrac{1}{1-\rho^2}\left[\left(\dfrac{ln\overline{q_c}q_c'-\mu_{qc}}{\sigma_{qc}}\right)^2 -\right.\right.$

$\left.\left. 2\rho\left(\dfrac{ln\overline{q_c}q_c'-\mu_{qc}}{\sigma_{qc}}\right)\left(\dfrac{ln\overline{q_r}q_r'-\mu_{qr}}{\sigma_{qr}}\right) + \left(\dfrac{ln\overline{q_r}q_r'-\mu_{qr}}{\sigma_{qr}}\right)^2\right]\right\},$

(6)

where $\sigma$ is standard deviation and $\rho$ is the correlation coefficient of $q_c$ and $q_r$.

Similarly, by integrating the accretion rate in Eq. (2) from Eq. (6), we get the accretion enhancement factor ($E_{accr}$) of:

$$E_{accr} = \left(1 + \frac{1}{v_{q_c}}\right)^{\frac{1.15^2 - 1.15}{2}} \left(1 + \frac{1}{v_{q_r}}\right)^{\frac{1.15^2 - 1.15}{2}} \exp(\rho 1.15^2 \sqrt{\ln\left(1 + \frac{1}{v_{q_c}}\right)\ln(1 + \frac{1}{v_{q_r}})}). \qquad (7)$$

**3. Ground-based observations and retrievals**

The datasets used in this study were collected at the Department of Energy (DOE)

Atmospheric Radiation Measurement (ARM) Mobile Facility (AMF), which was deployed on the northern coast of Graciosa Island (39.09$^\circ$N, 28.03$^\circ$W) from June 2009 to December

2010 (for more details, please refer to Rémillard et al. 2012; Dong et al. 2014a and Wood et al. 2015). The detailed operational status of the remote sensing instruments on AMF was summarized in Figure 1 of Rémillard et al. (2012) and discussed in Wood et al. (2015). The

ARM Eastern North Atlantic (ENA) site was established on the same island in 2013 and provides long-term continuous observations.

The cloud-top heights ($Z_{top}$) were determined from W-band ARM cloud radar (WACR)

reflectivity and only single-layered low-level clouds with $Z_{top} \leq 3$ km are selected. Cloud- base heights ($Z_{base}$) were detected by a laser ceilometer (CEIL) and the cloud thickness was simply the difference between cloud top and base heights. The cloud liquid water path (CLWP) was retrieved from microwave radiometer (MWR) brightness temperatures measured at 23.8 and 31.4 GHz using a statistical retrieval method with an uncertainty of 20

g m$^{-2}$ for CLWP < 200 g m$^{-2}$, and 10% for CLWP > 200 g m$^{-2}$ (Liljegren et al., 2001; Dong et al., 2000). Drizzling status is identified through a combination of WACR reflectivity and

Zbase. As in Wu et al. (2015), we label the status of a specific time as "drizzling" if the

WACR reflectivity below the cloud base exceeds -37 dBZ.

The ARM merged sounding data has1-min temporal and 20-m vertical resolution below 3

km (Troyan, 2012). In this study, the merged sounding profiles are averaged to 5-min resolution. Pressure and temperature profiles are used to calculate air density ($\rho_{air}$) profiles and to infer adiabatic cloud water content.

Cloud droplet number concentration ($N_c$) is retrieved using the methods presented in

Dong et al. (1998, 2014a and 2014b) and are assumed to be constant in a cloud layer

. Vertical profiles of cloud and rain water content (CLWC and

RLWC) are retrieved by combining WACR reflectivity, CEIL attenuated backscatter and by assuming adiabatic growth of cloud parcels. The detailed description is presented in

Appendix A with result from an example case. The CLWC and DLWC values are transformed to $q_c$ and $q_r$  by dividing by air density (e.g., $q_c(z) = CLWC(z)/\rho_{air}(z)$).

The estimated uncertainties for $q_c$ and $q_r$ retrievals are 30% and 18%, respectively (see

Appendix A). We used the range of $q_r$ and $q_c$ variations as inputs Eqs. (4) and (7) to assess the uncertainties in $E_{auto}$ and $E_{accr}$. For example, use $(1 \pm 0.3)q_c$ in Eq. (4) and the mean difference in $E_{auto}$ is then used as uncertainty. Same method is used to estimate the uncertainties for $E_{accr}$.

[revised manuscript text omitted]

Using LWP from the Moderate Resolution Imaging Spectroradiometer (MODIS), Wood and Hartmann (2006) found that cloud fraction decreases when the LWP field becomes more inhomogeneous and open cells are generally more inhomogeneous than closed cells. Open cells have been shown to be associated with stronger drizzling process (Comstock et al.,

2007). The relationship between reduced homogeneity and stronger precipitation intensity is found in this study, which is similar as those in Wood and Hartmann (2006), Comstock et.

(2007), among other studies (e.g., Barker et al., 1996; Pincus et al., 1999).

It is clear that  $q_c$ and $N_c$ in Figure 1b are correlated with each other. In addition to their natural relationships,  $q_c$ and $N_c$ in our retrieval method are also correlated (Dong et al. 2014a and 2014b). Thus, the effect of $q_c$ and $N_c$ covariance on $E_{auto}$ is not included in this study. In Figures 1c and 1d, the results are calculated using model grid of 180-km for the selected case on 27 July 2010. In Section 4.2, we will use these approaches to calculate their statistical results for multiple grid sizes using the 19-month ARM ground-based observations and retrievals.

**4.2 Statistical result**

For a specific grid size, e.g. 60-hour, we estimate the shape parameter ($\nu$)

and calculate $E_{auto}$ through Eqns. (5) and (7). The PDFs of $E_{auto}$ for both 60-km and

180-km grids are shown in Figures 2a-2d. The distributions of $E_{auto}$ values calculated from $q_c$ with 60-km and 180-km grids (Figures 2a and 2b) are  different to each other  (2.79 vs.

3.3). The calculated $E_{auto}$ values range from 1 to 10, and most are less than 4

. The average value for the 60-km grid (2.79) is smaller than that for the 180-km grid  (3.2), indicating a possible dependence of $E_{auto}$ on model grid size. Because drizzle-sized drops are  from the autoconversion , we investigate the relationship of $E_{auto}$ and precipitation frequency, which we define as the average percentage of drizzling occurrence based on radar reflectivity below cloud base.

Given the average LWP at Azores from Dong et al. (2014b, 109-140 g m$^{-2}$), the precipitation frequency (black lines in Figures 2a and 2b) agrees well with those from Kubar et al. (2009,

0.1-0.7 from their Figure 11). The precipitation frequency within each PDF bin shows an increasing trend for $E_{auto}$ from 0 to ~4-6, then oscillates around a relative constant when $E_{auto}$ > 6, indicating that in precipitation initiation process,

$E_{auto}$ keeps increasing to a certain value (~6) until the precipitation frequency reaches a near- steady state. Larger $E_{auto}$ values do not necessarily result in higher precipitation frequency but instead may produce more drizzle-sized drops from autoconversion process when the cloud is precipitating.

The PDFs of $E_{auto}$ calculated from $N_c$ also share similar patterns of positive skewness and peaks at ~1.5-2.0 for the 60-km and 180-km grids  (Figures 2c and 2d). Although the average values are close to their CLWP counterparts (2.54 vs. 2.79

for 60-km and 3.45 vs. 3.2 for 180-km), the difference between 60-km and 180-km grids becomes large. The precipitation frequencies within each bin do not show similar slightly decreasing trend like what is shown in Figures 2a and 2b. This suggests complicated effects of droplet number concentration on precipitation initiation and warrants more explorations of aerosol-cloud-precipitation interactions. This is very intriguing result, which suggests the existence of significant sub-grid variation of $N_c$ and this variation can significantly influence the warm rain process. As mentioned in Section 2, we also fit

$q_c$ and $N_c$ using lognormal distributions. The distributions of $E_{auto}$ are close to Figure 2

(not shown here) with average values of 3.28 and 3.84, respectively, for 60-km and 180- km grids. Because the $E_{accr}$ values calculated from $q_c$ and $N_c$ are close to each other, we will focus on analyzing the results from $q_c$ only for simplicity and clarity. The effect of $q_c$ and $N_c$ covariance, as stated in Section 4.1, is not presented in this study due to the intrinsic correlation in the retrieval (Dong et al., 2014a and 2014b and

Appendix A of this study).

The covariance of $q_c$ and $q_r$ is included in calculating $E_{accr}$ and the results are shown in Figures 2e and 2f. The calculated $E_{accr}$ values range from 1 to 4 with mean values of 1.62 and 1.76 for 60-km and 180-km grids

, respectively. These two mean values are much greater than the prescribed value used in  MG08 (1.07

). Since accretion  parameterizes the process  in which rain drops collect cloud droplets, we superimpose the ratio of $q_r$ to $q_c$ within each bin (black lines in

Figures 2e and 2f) to represent the portion of rain water in the cloud layer.

In both panels, the ratios are less than 15%, which means that $q_r$ can be more than one order of magnitude smaller than $q_c$. The differences in magnitude are consistent with previous

CloudSat and aircraft results (e.g., Boutle et al. 2014). This ratio increases from $E_{accr}$=0 to ~2, and then decreases, suggesting a possible optimal state for collision-coalescence process to achieve maximum efficiency for converting cloud water into rain water at $E_{accr}$=2. In other words, the conversion efficiency cannot be infinitely increased with $E_{accr}$ under fixed available cloud water. The ratios of $q_r$ to $q_c$ increases from $E_{accr}$=1.07 (0.063) to $E_{accr}$=2.0

(0.142), indicating that the fraction of rain water in total water is too low in the total water from the prescribed $E_{accr}$ and using a larger $E_{accr}$ value can increase this ratio, in other word, increase precipitation intensity. 
[revised manuscript text omitted]

---

## Referee Report (RR1)

Review of Wu et al., "Evaluation of autoconversion and accretion enhancement factors in GCM warm-rain parameterizations using ground-based measurements at the Azores," first revision

I must start by acknowledging the major effort undertaken by the authors to address my comments on the initial submission, especially with respect to obtaining collocated, vertically-resolved cloud and rain liquid water contents. I'm sure this was a significant undertaking, but it puts the premise of their analysis on solid footing and gives me much greater confidence in their conclusions. The addition of Appendix A to demonstrate the rain water content retrieval is also greatly appreciated. I still have a number of minor comments and there remain many language issues that introduce ambiguity in interpreting the authors' statements. I therefore recommend **acceptance pending minor revisions** to address these comments, and I **strongly** recommend the authors use a professional copyediting service to handle the language issues.

**Other general comments:**

- It's misleading that you call the effective length scale calculated from (wind speed * time elapsed) a "model grid size." It would be more accurate to use language such as "equivalent grid size" or "equivalent model grid size."
- There is a pervasive units issue. Almost everywhere that grid size is given as an area, the units are given in km where they should be $km^2$. Please correct throughout the manuscript or change all instances to "horizontal grid size of X km" such that they do not reference an area.
- Using a constant value of DSD width is analogous to using a constant LWC distribution width or $E_{auto}/E_{accr}$ width – the spectral width of the DSD is something that varies (increases) with length scale (e.g. Geoffroy et al, 2010, ACP) and is also dependent on the choice of in situ instrumentation (e.g. Witte et al., 2018, GRL; also mentioned but not dealt with in Miles et al., 2000, JAS). While I don't think there's an obvious "better" solution at this point (the math would get considerably hairier if one also considers variable DSD shape), I think it's important to note that by making this assumption, you're essentially just kicking the use of "parameters constant with length scale" down the line from LWC distributions to DSD width. This seems particularly prescient given that your argument for regime-dependence is reflected in the finding of Miles et al. (2000) that the distribution of DSD parameters is different for marine and continental clouds.

**Specific content-related comments** are given in the remainder of the review in reference to page and line number(s) (format: PX, LY-Z = page X, lines Y-Z). Please see the annotated PDF following these comments for a non-comprehensive list of language issues.

P1, L33: This sentence gives the impression that there is a particular value that the $q_r$-$q_c$ ratio *should* be. Be more explicit here about why this value matters – just throwing out the values without context (e.g. observed vs. typical GCM values) doesn't "prove" that the presently-used constant enhancement factors are wrong.
P3, L55: How do aerosol effects being tied to precipitation suppression fit with the rest of your argument? I fail to see the relevance, especially since your results show that precipitation frequency has almost no relationship with $E_{auto,Nc}$.

P4, L60-62: I don't see the point of this sentence. If you're getting at the idea that the "cloud" and "rain" sub-categories are an arbitrary division by drop size, say so.

P4, L72: Is there not one single study that parameterizes accretion as something other than a power law? You're on firmer ground saying "The vast majority" or something like that because it only takes one counterexample to make your statement false.

P5, L91-93: Cheng and Xu's autoconversion equation is independent of vertical velocity and rain mixing ratio (Text in right column of pg 2319 and their Eq. 6).

P6, L 109: Why is recognition of spatial scales the important aspect? This seems trivial. Please expand.

P6, L113-114: Do you mean Zhang et al. derived sub-grid distributions or the actual pixel-scale CLWP and Nc? Please clarify.

P7, L131-132: Enhancement factors are applied at the grid scale by definition, so it seems repetitive to say "grid-mean process enhancement factors."

P15, L268: You don't show $q_r$ in Fig 1. Either reference rain LWC as shown in Fig. A1 or remove "and $q_r$" from the sentence.

P17, L305: What does "relative constant" mean? Clarify what the adjective means or remove it.

P18, L317-319: There are a number of other explanations for this result: the dependence of autoconversion parameterizations on number concentration may be flawed, there could be problems with your number concentration retrieval, or perhaps the assumption of constant $N_c$ with height doesn't work. Unless you have evidence that a) there is significant subgrid variability of $N_c$ and b) it matters at the process level, I don't think you can make this statement.

P22, L402-404: The differences from prescribed values only have explanatory power if the simulations used to diagnose it were run at something comparable to 30 km horizontal resolution. Do you have a reference to support this?

P22, L410: The language "...the location we choose to collect ground-based observations..." implies that the authors were the primary decision makers regarding the location of the CAP-MBL deployment. If this is the case, this wording is fine. Otherwise, considering alternate wording, e.g. "...the location of the ground-based observations and retrievals used in this study is..."
P25, L464: Use the $180^2$ $km^2$ value here for consistency. You use $120^2$ $km^2$ here while you show a maximum grid size of $180^2$ $km^2$ in the figures and use this same maximum grid size in reference to $E_{accr}$ below.

P28, L519-520: This sentence is confusing. I think you're trying to say "if the vertical gradient of $N_w$ is negative below cloud base" - can you confirm?

P29, L526-527: Italicize all instances in the text of $Z$ ($Z_c$, $Z_d$, etc.). Also, does the subscript "d" indicate drizzle? You use the subscript "r" in Eq. A1 and elsewhere in the main manuscript.

P29, L530: Rearrange Eq. A2 for the value you're actually solving for (CLWC(1)_reflectivity).

P29, L536: Eq. A2 only gives reflectivity at cloud base. How do you integrate up for the profile?

P30, L543-544: What if there is no drizzle at cloud top? How good is the assumption that you can just decrease $N_W$ until your criteria is satisfied?

Table 2 (P 42, L819): While visually repetitive, the table is more readable if the upper left corner cell is clear. The entries in the leftmost column should then read (from top to bottom): LTS > 18 K, 13.5 < LTS < 18 K, LTS <13.5 K. You may also consider placing a vertical line between the first and second columns to differentiate between the category column and results/data.

Figure 1, panel c (P44, L825): Why does the histogram look so much different than the fitted gamma distribution? Assuming 5-10 m/s wind speeds there would be something like 60-120 samples for an equivalent 60 km scale, so is this just a consequence of coarse binning?

Figure 1 caption (P44, L 832): Replace "mean-$q_c$" with an overbar over $q_c$.

Figure 2, panels e-f (P45, L 839): The label "cov(qc, qr)" is misleading because you don't actually show the covariance anywhere.

Figure 2, panels e-f (P45, L839): Consider adding minor ticks to the y axes or plot the ratios on a separate right axis. It's very difficult to get a sense for the maximum magnitude of qr/qc because it's all below the 0.2 tick.

Figure 2 (P45, L839): Add labels to the y axes to show that both the PDF and (precip frequency/qr-qc ratio) are shown on the same scale, i.e. label y axes "PDF, precipitation frequency" and "PDF, qr-qc ratio" or similar. This information shouldn't be buried at the bottom of the caption.

Figure 5 (P48, L856): How do these adjustments compare with retrieval uncertainty? I ask because you show in Fig. 4 that uncertainty in E decrease with equivalent grid size. For example, if the retrieval uncertainty at 30 km is 20%, then does a $q_c$ adjustment tell you something physical or is it just another way of describing uncertainty?

[revised manuscript text omitted]

---

## Editor Decision (ED1)

[revised manuscript text omitted]

*[Handwritten margin notes: "Frisch et al. 95 used -17 dBZ. Isn't -37 a very strong threshold?" and "Please explain what cloud types you analyse. How much is SCu vs. Cu?"]*

The ARM merged sounding data have a 1-min temporal and 20-m vertical resolution below

3 km (Troyan, 2012). In this study, the merged sounding profiles are averaged to 5-min resolution. Pressure and temperature profiles are used to calculate air density ($\rho_{air}$) profiles and to infer adiabatic cloud water content.

Cloud droplet number concentration ($N_c$) is retrieved using the methods presented in Dong et al. (1998, 2014a and 2014b) and are assumed to be constant in a cloud layer. Vertical profiles of cloud and rain water content (CLWC and RLWC) are retrieved by combining WACR

reflectivity, CEIL attenuated backscatter and by assuming adiabatic growth of cloud parcels.

The detailed description is presented in Appendix A with the results from a selected case. The

CLWC and RLWC values are transformed to $q_c$ and $q_r$ by dividing by air density (e.g., $q_c(z) =$

$CLWC(z)/\rho_{air}(z)$).

The estimated uncertainties for the retrieved $q_c$ and $q_r$ are 30% and 18%, respectively (see

Appendix A). We used the estimated uncertainties of $q_r$ and $q_c$ as inputs of Eqs. (4) and (7) to assess the uncertainties of $E_{auto}$ and $E_{accr}$. For instance, $(1 \pm 0.3)q_c$ are used in Eq. (4) and the mean differences are then used as the uncertainty of $E_{auto}$. Same method is used to estimate the uncertainty for $E_{accr}$.

The autoconversion and accretion parameterizations partitioned from the collision- coalescence process dominate at different levels in a cloud layer. Autoconversion dominates around cloud top where cloud droplets reach maximum by condensation and accretion is

*handwritten notes:* Do you mean constant with height?

water

A

*

The

* not appropriate for Cumulus

Auto conversion is the process of self

*[handwritten top margin: collection of cloud droplets !!]*

[revised manuscript text omitted]

---

## Author Response (AR2)

**A Point-by-point Response to Review Comments**

Dear Dr. Feingold,

We are submitting the revised manuscript (#acp-2018-499) for your consideration of publication in *Atmospheric Chemistry and Physics*. We have carefully studied the reviewer's comments and revised the manuscript accordingly. Please find the point-by-point response (marked as blue) to the review comments. We have provided a copy of track-change manuscript as well as a clean copy of the revised manuscript.

Thank you for your consideration of this submission. We hope you find our response adequately address the review comments and the revision acceptable. We would greatly appreciate it if you could get back to us with your decision at your earliest convenience.

Sincerely,

Peng Wu

Department of Hydrology and Atmospheric Sciences

University of Arizona

Tucson, AZ 85721, USA

**Review of Wu et al., "Evaluation of autoconversion and accretion enhancement factors in GCM warm-rain parameterizations using ground-based measurements at the Azores," first revision**

I must start by acknowledging the major effort undertaken by the authors to address my comments on the initial submission, especially with respect to obtaining collocated, vertically-resolved cloud and rain liquid water contents. I'm sure this was a significant undertaking, but it puts the premise of their analysis on solid footing and gives me much greater confidence in their conclusions. The addition of Appendix A to demonstrate the rain water content retrieval is also greatly appreciated. I still have a number of minor comments and there remain many language issues that introduce ambiguity in interpreting the authors' statements. I therefore recommend **acceptance pending minor revisions** to address these comments, and I **strongly** recommend the authors use a professional copyediting service to handle the language issues.

Thanks for the insightful comments and suggestions for this and the initial submission, which helped to improve the manuscript a lot.
We have made point-to-point revisions and did a thorough proofreading for the revised manuscript.

**Other general comments:**
- It's misleading that you call the effective length scale calculated from (wind speed * time elapsed) a "model grid size." It would be more accurate to use language such as "equivalent grid size" or "equivalent model grid size."
- There is a pervasive units issue. Almost everywhere that grid size is given as an area, the units are given in km where they should be km2. Please correct throughout the manuscript or change all instances to "horizontal grid size of X km" such that they do not reference an area.
Thanks for the suggestions.
We have changed the terminology to 'horizontal equivalent grid size'. For simplicity and convenience, it is referred to as 'equivalent size' in the text.

- Using a constant value of DSD width is analogous to using a constant LWC distribution width or $E_{auto}/E_{accr}$ width – the spectral width of the DSD is something that varies (increases) with length scale (e.g. Geoffroy et al, 2010, ACP) and is also dependent on the choice of in situ instrumentation (e.g. Witte et al., 2018, GRL; also mentioned but not dealt with in Miles et al., 2000, JAS). While I don't think there's an obvious "better" solution at this point (the math would get considerably hairier if one also considers variable DSD shape), I think it's important to note that by making this assumption, you're essentially just kicking the use of "parameters constant with length scale" down the line from LWC distributions to DSD width. This seems particularly prescient given that your argument for regime-dependence is reflected in the finding of Miles et al. (2000) that the distribution of DSD parameters is different for marine and continental clouds.
Thanks for the comment.
The assumed DSD width is used to retrieve LWC within a radar range gate and it is important to note that the LWC within a radar range gate is a bulk property and do not have a distribution. The qc distribution within a specific time interval is used in the Gamma fitting and this distribution do not have a constant width and is little affected by the constant with od DSD. However, we agree with the reviewer that DSD width should vary with different sampling length/time and different instruments. We added the following discussion in Append A to feather this: 'Geoffroy et al. (2010) suggested that $\sigma_x$ increases with the length scale and Witte et al.

(2018) showed that $\sigma_x$ also dependent on the choice of instrumentation. The variations of $\sigma_x$ should be reflected in the retrieval by using different $\sigma_x$ values with time. However, no aircraft measurements were available during CAP-MBL to provide $\sigma_x$ over the Azores region. The inclusion of solving $\sigma_x$ in the retrieval adds another degree of freedom to the equations and complicates the problem considerably. In this study, $\sigma_x$ is set to a constant value of 0.38 from Miles et al. (2000), which is a statistical value from aircraft measurements of marine low-level clouds.'

**Specific content-related comments** are given in the remainder of the review in reference to page and line number(s) (format: PX, LY-Z = page X, lines Y-Z). Please see the annotated PDF following these comments for a non-comprehensive list of language issues.
P1, L33: This sentence gives the impression that there is a particular value that the qr-qc ratio should be. Be more explicit here about why this value matters – just throwing out the values without context (e.g. observed vs. typical GCM values) doesn't "prove" that the presently-used constant enhancement factors are wrong.
Thanks for the comment.
This sentence has been revised to 'The ratios of rain to cloud water mixing ratio at $E_{accr}$=1.07 and $E_{accr}$=2.0 are 0.063 and 0.142, respectively, from observations, further suggesting that the prescribed value'.

P3, L55: How do aerosol effects being tied to precipitation suppression fit with the rest of your argument? I fail to see the relevance, especially since your results show that precipitation frequency has almost no relationship with $E_{auto,Nc}$.
Thanks for the comment. This sentence has been deleted.

P4, L60-62: I don't see the point of this sentence. If you're getting at the idea that the "cloud" and "rain" sub-categories are an arbitrary division by drop size, say so.
Thanks for the comment.
This sentence is an elaboration of the parameterized rather than resolved process in GCMs and uses warm rain process as an example. We do not intend to explain the separation of rain and cloud sub-categories here.

P4, L72: Is there not one single study that parameterizes accretion as something other than a power law? You're on firmer ground saying "The vast majority" or something like that because it only takes one counterexample to make your statement false.
Thanks for the comment.
This sentence has been rephrased in the revised manuscript.

P5, L91-93: Cheng and Xu's autoconversion equation is independent of vertical velocity and rain mixing ratio (Text in right column of pg 2319 and their Eq. 6).
Thanks for the comment. This sentence has been revised.

P6, L 109: Why is recognition of spatial scales the important aspect? This seems trivial. Please expand.
Thanks for the comment.

Because the inhomogeneity parameter, similar as the enhancement factor in this study, characterizes cloud field in a grid. An inhomogeneous cloud in a GCM grid can be homogeneous in a WRF grid. Also, in the lon-lat grid setting GCMs the grid size can be dramatically different in tropical and polar regions. The inhomogeneity parameter in Xie and Zhang (2015) can be applied to the globe without manually tuning.

We expanded the following in the revised manuscript: "…found that it can recognize spatial scales without manual tuning and can be applied to the entire globe".

P6, L113-114: Do you mean Zhang et al. derived sub-grid distributions or the actual pixel-scale CLWP and Nc? Please clarify.

Thanks for the comment.

Zhang et al. (2018) derived sub-grid distributions of CLWP and Nc and describe them using lognormal and gamma distributions. This is clarified in the revised manuscript.

P7, L131-132: Enhancement factors are applied at the grid scale by definition, so it seems repetitive to say "grid-mean process enhancement factors."

Thanks for the comment.

This has been rephrased to 'enhancement factors' in the revised manuscript.

P15, L268: You don't show qr in Fig 1. Either reference rain LWC as shown in Fig. A1 or remove "and qr" from the sentence.

Thanks for the comment.

This sentence has been rephrased to '…as seen from WACR reflectivity and $q_r$ in Figure A1.' in the revised manuscript.

P17, L305: What does "relative constant" mean? Clarify what the adjective means or remove it.

Thanks for the comment. This has been removed.

P18, L317-319: There are a number of other explanations for this result: the dependence of autoconversion parameterizations on number concentration may be flawed, there could be problems with your number concentration retrieval, or perhaps the assumption of constant Nc with height doesn't work. Unless you have evidence that a) there is significant subgrid variability of Nc and b) it matters at the process level, I don't think you can make this statement.

Thanks for the comment. We decided to delete this statement in the revised manuscript.

P22, L402-404: The differences from prescribed values only have explanatory power if the simulations used to diagnose it were run at something comparable to 30 km horizontal resolution. Do you have a reference to support this?

Thanks for the comment.

The problem of 'too frequent and too light' is most prominent for the equivalent size of 30-km from the aspect of enhancement factors. We are not aware of any reference that draw similar conclusion.

P22, L410: The language "…the location we choose to collect ground-based observations…" implies that the authors were the primary decision makers regarding the location of the CAP-MBL deployment. If this is the case, this wording is fine. Otherwise, considering alternate wording, e.g. "…the location of the ground-based observations and retrievals used in this study is…"

Thanks for the comment. This sentence has been re-written in the revised manuscript.

P25, L464: Use the $180^2$ km$^2$ value here for consistency. You use $120^2$ km$^2$ here while you show a maximum grid size of $180^2$ km$^2$ in the figures and use this same maximum grid size in reference to Eaccr below.

Thanks for the comment.

This sentence has been changed in the revised to 'The calculated $E_{auto}$ values from observations and retrievals increase from 1.96 at an equivalent size of 30 km to 3.18 at an equivalent size of 150 km. These values are 38% and 0.625% lower than the prescribed value of 3.2. The prescribed value in MG08 represents well in large grid sizes in GCMs (e.g., $180^2$ km$^2$ grid).' in the revised manuscript.

P28, L519-520: This sentence is confusing. I think you're trying to say "if the vertical gradient of Nw is negative below cloud base" - can you confirm?

Thanks for the comment.

Yes, if the Nw vertical gradient below cloud base is negative, we use constant Nw in cloud.

This sentence has been rephrased in the revised manuscript.

P29, L526-527: Italicize all instances in the text of Z (Zc, Zd, etc.). Also, does the subscript "d" indicate drizzle? You use the subscript "r" in Eq. A1 and elsewhere in the main manuscript.

Thanks for the comment. These have been changed in the revised manuscript.

P29, L530: Rearrange Eq. A2 for the value you're actually solving for (CLWC(1)_reflectivity).

Thanks for the comment. The equation has been rearranged.

P29, L536: Eq. A2 only gives reflectivity at cloud base. How do you integrate up for the profile?

Thanks for the comment. The reflectivity is calculated from the updated adiabatic LWC (after multiplying by $s$). for clarification, the sentence is rephrased as 'Reflectivity profile from cloud is then calculated from Eq. (A2.1) using the updated $CLWC_{adiabatic}$'.

P30, L543-544: What if there is no drizzle at cloud top? How good is the assumption that you can just decrease NW until your criteria is satisfied?

Thanks for the comment. The big assumption in the rain estimation method is that, whenever rain occurs below cloud base, it exists in the whole cloud layer. This assumption may not hold in situations that the top layer is affected by entrainment and no rain drops exist. However, without in situ measurement, we are unable to identify if rain drops exist near cloud in a case by case base.

The minimum value of Nw at the top layer in our estimation is on the order of $10^{-4}$, which is considered a good approximation to the situation that Nw is zero and this approximation has little effect on the reflectivity calculation.

Table 2 (P 42, L819): While visually repetitive, the table is more readable if the upper left corner cell is clear. The entries in the leftmost column should then read (from top to bottom): LTS > 18

K, 13.5 < LTS < 18 K, LTS <13.5 K. You may also consider placing a vertical line between the first and second columns to differentiate between the category column and results/data.
Thanks for the comment. Table 2 has been changed in the revised manuscript.

Figure 1, panel c (P44, L825): Why does the histogram look so much different than the fitted gamma distribution? Assuming 5-10 m/s wind speeds there would be something like 60-120 samples for an equivalent 60 km scale, so is this just a consequence of coarse binning?
Thanks for catching this. It was a mistake in our plotting code. The figure has been updated in the revised manuscript.

Figure 1 caption (P44, L 832): Replace "mean-qc" with an overbar over qc.
Thanks, this is changed.

Figure 2, panels e-f (P45, L 839): The label "cov(qc, qr)" is misleading because you don't actually show the covariance anywhere.
Figure 2, panels e-f (P45, L839): Consider adding minor ticks to the y axes or plot the ratios on a separate right axis. It's very difficult to get a sense for the maximum magnitude of qr/qc because it's all below the 0.2 tick.
Figure 2 (P45, L839): Add labels to the y axes to show that both the PDF and (precip frequency/qr-qc ratio) are shown on the same scale, i.e. label y axes "PDF, precipitation frequency" and "PDF, qr-qc ratio" or similar. This information shouldn't be buried at the bottom of the caption.
Thanks for the comment. We have changed cov(qc, qr) to qc, qr, changed y-axis label and added minor ticks in the revised manuscript.

Figure 5 (P48, L856): How do these adjustments compare with retrieval uncertainty? I ask because you show in Fig. 4 that uncertainty in E decrease with equivalent grid size. For example, if the retrieval uncertainty at 30 km is 20%, then does a qc adjustment tell you something physical or is it just another way of describing uncertainty?
Thanks for the comment.
We do not think the percentages in Figure 5c is a direct result of retrieval uncertainty. Rather, the percentages in Figures 5a and 5b may result from uncertainties of qc retrieval. For example, for the 60-km equivalent size, the broad range from -40% to 40% is the actual adjustment and is greater than the retrieval error.
The averaged adjustment for 180-km equivalent size is zero but the uncertainty in Eaccr is comparable to the 60-km equivalent size.
This figure should be considered as another way of demonstrating the information in Figure 4. If the retrieval uncertainty is to be considered, a similar shaded area as in Figure 4 should be around the solid line in Figure 5c.

[revised manuscript text omitted]

---

## Author Response (AR3)

**A Response to Editor Comments**

Dear Dr. Feingold,

Thank you for the corrections and comments on our manuscript (#acp-2018-499). They are greatly appreciated.

We are submitting a revised manuscript for your consideration of publication in *Atmospheric Chemistry and Physics*. We have carefully studied your comments and revised the manuscript accordingly. Please find the response (marked as blue) to the major comments. We have provided a copy of track-change manuscript as well as a clean copy of the revised manuscript.

Thank you for your consideration of this submission. We hope you find our response adequately address your comments and the revision/corrections acceptable. We would greatly appreciate it if you could get back to us with your decision at your earliest convenience.

Sincerely,

Peng Wu

Department of Hydrology and Atmospheric Sciences

University of Arizona

Tucson, AZ 85721, USA

Lines 62-65, 220-223: definition and understanding of autoconversion and accretion.

Thanks for the comments and corrections.

We were trying to say, '*cloud* droplets reach maximum size by *condensation* near cloud top' and autoconversion is dominant at the top part of the cloud layer where cloud droplets collide to form drizzle drops.

To avoid confusion, the definition and interpretation in lines 62-65 and 220-223 have been rephrased:

Lines 62-65 (62-64 in the revision): 'Autoconversion represents the process of drizzle drops being formed through the self-collection of cloud droplets and accretion represents the process where rain drops grow by collecting cloud droplets.'

Lines 220-223 (228-230 in the revision): 'Autoconversion dominates around cloud top where drizzle drops form by the self-collection of cloud droplets and accretion is dominant at middle and lower parts of the cloud where rain drops grow by collecting cloud droplets.'

Lines 193-194: clarify cloud types; line 210: validity of 'adiabatic growth' assumption.

Thanks for this comment.

In line 194 of the revised manuscript, the cloud selection criteria were clarified: 'only single-layered and overcast low-level clouds with $Z_{top} \leq 3$ km were selected'.

In the retrieval method in Append A, the layer-mean $N_c$ is from Dong et al. (2014a and 2014b), in which only singled layered and overcast low-level cloud properties were retrieved. Therefore, the analysis in this study is for stratus and stratocumulus, which makes the adiabatic assumption appropriate when retrieving cloud liquid water content. No cumulus properties are retrieved in this study.

Line 202: reflectivity threshold to identify drizzling clouds

Thank you for the comment.

We added the following to lines 203-210 in the revised manuscript to comment on the reflectivity threshold.

'Note the differences of the reflectivity thresholds used here and in other studies. For example, -15 dBZ in Sauvageot and Omar (1987), -17 dBZ in Frisch et al. (1995), -19 to -16 dBZ in Wang and Geerts (2003) and -30 dBZ or lower in Kollias et al. (2011). The threshold used in this study is set at the cloud base rather than for the entire cloud layer as in the abovementioned studies. The -37 dBZ threshold is a statistical value from WACR observations at the Azores presented by Wu et al. (2015, Figure 2a), in which it is found that using a higher threshold will miss a significant number of drizzling events especially the clouds with virga.'

Lines 517, 531: lognormal distribution for cloud DSD and normalized gamma distribution for rain DSD.

In the retrieval method, cloud drop size distribution (DSD) is assumed to be lognormal distribution and rain DSD is assumed to be normalized gamma distribution, which are the common practice when retrieving cloud (e.g., Frisch et al., 1995; Dong et al., 1997; McFarlane et al., 2003; Fielding et al., 2015) and rain (e.g., O'Connor et al., 2005; Fielding et al., 2015; Posselt et al., 2017) microphysical properties.

The parameter from the retrieval we used is the cloud liquid water content (CLWC) rather than DSD. Frisch et al. (1998) suggested that, given radar reflectivity and a constraint on total liquid water path (LWP), the retrieved CLWC is relatively insensitive to the assumptions about size distribution.

In this study, to be consistent with the Morrison and Gettleman (2008) scheme, the spatial distribution (approximated using temporal distribution) of $q_c$ is fitted using gamma distribution. The lognormal fitting was also performed, and the results are very similar to those from gamma fitting (lines 325-328 in the revision).

[revised manuscript text omitted]